# SCF$^{Fbxw5}$ targets kinesin-13 proteins to facilitate ciliogenesis

Jörg Schweiggert[1],[*] (ID), Gregor Habeck[1] (ID), Sandra Hess[2],[3] (ID), Felix Mikus[1] (ID), Roman Beloshistov[1] (ID), Klaus Meese[1], Shoji Hata[1] (ID), Klaus-Peter Knobeloch[2] (ID) & Frauke Melchior[1],[**] (ID)

## Abstract

Microtubule depolymerases of the kinesin-13 family play important roles in various cellular processes and are frequently overexpressed in different cancer types. Despite the importance of their correct abundance, remarkably little is known about how their levels are regulated in cells. Using comprehensive screening on protein microarrays, we identified 161 candidate substrates of the multi-subunit ubiquitin E3 ligase SCF$^{Fbxw5}$, including the kinesin-13 member Kif2c/MCAK. *In vitro* reconstitution assays demonstrate that MCAK and its closely related orthologs Kif2a and Kif2b become efficiently polyubiquitylated by neddylated SCF$^{Fbxw5}$ and Cdc34, without requiring preceding modifications. In cells, SCF$^{Fbxw5}$ targets MCAK for proteasomal degradation predominantly during $G_2$. While this seems largely dispensable for mitotic progression, loss of Fbxw5 leads to increased MCAK levels at basal bodies and impairs ciliogenesis in the following $G_1/G_0$, which can be rescued by concomitant knockdown of MCAK, Kif2a or Kif2b. We thus propose a novel regulatory event of ciliogenesis that begins already within the $G_2$ phase of the preceding cell cycle.

**Keywords** cilia; Cullin-RING ligase; Fbxw5; MCAK; ubiquitin
**Subject Categories** Cell Adhesion, Polarity & Cytoskeleton; Cell Cycle; Post-translational Modifications & Proteolysis
The EMBO Journal (2021) 40: e107735

## Introduction

In the past decade, primary cilia have gained massive attention due to the discovery of their involvement in a plethora of human diseases, ranging from organ-related disorders such as polycystic kidney disease to more pleiotropic syndromes like Bardet–Biedl (Ansley *et al*, 2003; Lin *et al*, 2003; Reiter & Leroux, 2017). By forming antenna-like membrane protrusions that enrich diverse cellular receptors, primary cilia are important signalling hubs for tissue development and homeostasis (Malicki & Johnson, 2017).

Ciliogenesis takes place in almost all human cells and requires drastic remodelling of centrosomes that includes their migration to the cortex and a microtubule-dependent generation of a plasma membrane protrusion (Sánchez & Dynlacht, 2016; Werner *et al*, 2017). Upon re-entry into the cell cycle, primary cilia must be resorbed in order to release centrosomes for spindle formation and thus reappear only after mitotic exit (Izawa *et al*, 2015).

Recently, the ubiquitin–proteasome system (UPS) has been identified as an important regulator of ciliogenesis (Shearer & Saunders, 2016; Hossain & Tsang, 2019; Boukhalfa *et al*, 2019). Ubiquitylation defines the attachment of ubiquitin to substrate proteins via an enzymatic cascade, in which the final transfer to target lysine residues is catalysed by a so-called E3 ligase in a highly specific manner (Ciehanover *et al*, 1978; Kerscher *et al*, 2006; Zheng & Shabek, 2017). Since ubiquitin itself contains acceptor lysine residues for other ubiquitin moieties, chains of different linkages can be formed, some of which lead to proteasomal degradation (Kwon & Ciechanover, 2017). Examples for ubiquitin-dependent control of ciliogenesis comprise the E3 ligase Mindbomb1 that antagonises ciliogenesis by targeting Talpid3 and the Cullin-RING ligases (CRL) Cul3-KCTD17 and SCF$^{Fbxw7}$ that promote ciliogenesis via degradation of trichoplein and Nde1, respectively (Kasahara *et al*, 2014; Maskey *et al*, 2015; Wang *et al*, 2016).

Cullin-RING ligases constitute the biggest family of ubiquitin E3 ligases being responsible for up to 20% of proteolytic ubiquitylation events in cells (Petroski & Deshaies, 2005; Soucy *et al*, 2009). The best studied class of CRLs is SCF (Skp1-Cul1-F-box protein) E3 ligases that use Cul1 as a central scaffold protein. Cul1 recruits a small RING (really interesting new gene) containing protein called Rbx1 via its C-terminus and a substrate receptor module composed of Skp1 and one out of 69 interchangeable F-box proteins via its N-terminus (Lydeard *et al*, 2013; Skaar *et al*, 2013). Catalytic activity of SCF E3 ligases requires the modification of Cul1 with the ubiquitin-like modifier Nedd8. This induces conformational changes that facilitate ubiquitin transfer from the Rbx1-bound E2-Ubiquitin thioester to the substrate recruited by the F-box protein (Duda *et al*, 2011; Baek *et al*, 2020). Fbxw5 belongs to the WD40-domain containing family of F-box proteins. So far, known substrates of SCF$^{Fbxw5}$ include the actin remodeller Eps8, the centrosomal scaffold

1 Zentrum für Molekulare Biologie der Universität Heidelberg (ZMBH), University of Heidelberg, DKFZ - ZMBH Alliance, Heidelberg, Germany
2 Institute of Neuropathology, Faculty of Medicine, University of Freiburg, Freiburg, Germany
3 Faculty of Biology, University of Freiburg, Freiburg, Germany
*Corresponding author. Tel: +49 6221 54 6831; E-mail: joerg.schweiggert@gmail.com
**Corresponding author. Tel: +49 6221 54 6831; E-mail: f.melchior@zmbh.uni-heidelberg.de

protein Sas6, the COPII component Sec23b and apoptosis signal-regulating kinase 1 (Ask1) (Puklowski *et al*, 2011; Werner *et al*, 2013; Jeong *et al*, 2018; Bai *et al*, 2019). In addition, Fbxw5 has been suggested to be also active within a Cul4A/DDB1-containing CRL, in which it targets the tumour suppressors DLC1 and Tsc2 (Hu *et al*, 2008; Kim *et al*, 2013).

In order to discover further substrates of Fbxw5, we now employed comprehensive substrate screening on protein microarrays and identified Kif2c/MCAK (mitotic centromere-associated kinesin) as an important novel target of SCF$^{Fbxw5}$. MCAK is the best studied member of the kinesin-13 family, which are non-motile kinesins that use their central motor domain to depolymerise microtubules (MTs) in an ATP-dependent manner (Wordeman & Mitchison, 1995; Hunter *et al*, 2003; Friel & Welburn, 2018). MCAK is mostly known for its role in mitosis, where it localises to spindle poles, spindle MTs and centromeres exerting crucial roles in spindle formation and chromosome segregation (Walczak *et al*, 1996; Maney *et al*, 1998; Kline-Smith *et al*, 2004). Its activity is heavily regulated through phosphorylation by different mitotic kinases, such as AuroraA, AuroraB, Cdk1 and Plk1 (Andrews *et al*, 2004; Zhang *et al*, 2008, 2011; Sanhaji *et al*, 2010; Ems-McClung *et al*, 2013). Recently, MCAK has been implicated also in non-mitotic events, such as DNA damage repair and ciliogenesis (Miyamoto *et al*, 2015; Zhu *et al*, 2020).

Here, we show that SCF$^{Fbxw5}$ regulates MCAK protein levels by specifically catalysing K48 ubiquitin chains on MCAK in a highly efficient manner. Although this process occurs predominantly during G$_2$/M, loss of Fbxw5 does not provoke severe defects in mitosis but instead impairs ciliogenesis later in the following G$_0$ phase. Our work thus suggests an intriguing regulatory process required for ciliogenesis that seems to take place not during the event of ciliogenesis itself but rather within the preceding cell cycle.

# Results

### *In vitro* ubiquitylation screening identifies 161 candidate substrates of SCF$^{Fbxw5}$

In order to identify novel substrates of the SCF$^{Fbxw5}$ complex, we performed an *in vitro* ubiquitylation screen on commercial protein microarrays (ProtoArray® v5.0, Thermo Fisher Scientific) containing more than 9,000 human proteins expressed and purified as GST-fusion proteins from insect cells. Conditions that we have previously used for *in vitro* ubiquitylation of the SCF$^{Fbxw5}$ substrate Eps8 served as a blueprint for the screen (Werner *et al*, 2013). Two arrays were incubated with recombinant E1, E2s (UbcH5b and Cdc34 together), Ubiquitin, neddylated SCF$^{Fbxw5}$ complex (generated by a split and co-express method (Li *et al*, 2005)) and an ATP regenerating system to ensure reproducibility (Fig 1A). As a control, two arrays were probed with the same mix lacking SCF complexes. To detect ubiquitylated proteins, arrays were stringently washed, incubated with FK2 antibodies specific for conjugated ubiquitin (Fujimuro & Yokosawa, 2005), scanned, quantified and normalised (Fig 1B). As shown in Fig EV1A, duplicate reactions were highly reproducible. SCF$^{Fbxw5}$-specific ubiquitylation substrates were then selected based on two criteria: first, signal intensity above an arbitrary threshold (i.e. 500 AU) and second, a more than 5-fold increase in signal

intensity over control arrays lacking E3 ligase (Figs 1C and EV1B). We identified a total of 161 candidates (Dataset EV1) fulfilling the above-mentioned criteria. Among them were the previously identified substrates Sas6 and Sec23b, demonstrating the capability of the *in vitro* system to identify SCF$^{Fbxw5}$ targets (Fig 1C, note that Ask1 and Eps8 were not present on the array). Gene ontology enrichment analysis of cellular components via the DAVID webtool (Huang *et al*, 2009a, 2009b) revealed that a high proportion of these substrates localise to the cytoplasm, which is in line with the described cytoplasmic localisation of Fbxw5 (Puklowski *et al*, 2011). Furthermore, components of the cytoskeleton and vesicle-based transport were enriched among the SCF$^{Fbxw5}$ substrates (Fig 1D). This fits well to the previously identified substrates Eps8, Sas8 and Sec23b and suggests that Fbxw5 could act as a master regulator of cytoskeletal-dependent transport processes. In order to further validate our screen, we picked candidates from these categories and tested their ubiquitylation efficiency by SCF$^{Fbxw5}$ in solution with purified substrates. All selected candidates were efficiently ubiquitylated (Fig 1E) and a high proportion of these substrates interacted with Fbxw5 in co-immunoprecipitation (co-IP) experiments (Fig EV1C and D), demonstrating the high reliability of our screen.

### Fbxw5 interacts with kinesin-13 family members

Due to its pivotal role in different cellular processes and its strong signal intensity in the screen, we focused on the candidate Kif2c/MCAK for further studies. Since ubiquitylation by SCF complexes requires substrate recruitment, we first tested whether MCAK interacts with the substrate receptor Fbxw5. As shown by co-IP of tagged proteins, MCAK binds to full length and an F-box lacking mutant of Fbxw5, but not to Fbxw7, indicating an interaction that is independent of other SCF components (Fig 2A). Accordingly, *in vitro* pull-down experiments using purified proteins mixed with competing *Escherichia coli* lysates revealed stoichiometric precipitation of MCAK with the Fbxw5/Skp1 sub-complex with no other specific proteins present in the pull-down, confirming that MCAK binds Fbxw5 in a direct and efficient manner (Fig 2B). In order to test whether MCAK binds Fbxw5 also in intact cells, we carried out NanoBRET (Machleidt *et al*, 2015) experiments by overexpressing HaloTag(HT)-tagged Fbxw5 and MCAK fused to NanoLuc luciferase in HeLa cells. Here, Fbxw5 and MCAK generated significantly stronger signals than the negative controls (Fig 2C), confirming that both proteins are also able to interact in living cells. Finally, we tested whether MCAK interacts with Fbxw5 at endogenous levels and carried out co-IP experiments using Fbxw5-directed antibodies under different cell cycle arrest conditions (Fig 2D). Interestingly, endogenous MCAK efficiently co-precipitated with Fbxw5 only upon nocodazole treatment. This may indicate an enhanced interaction during mitosis but it also could simply be due to increased levels of MCAK under these arrest conditions. In any case, this experiment confirms that both proteins are able to interact also at endogenous levels.

Human cells express two orthologs of MCAK—Kif2a and Kif2b—that have some overlapping and non-overlapping functions in cells (Manning *et al*, 2007; Welburn & Cheeseman, 2012). Kif2a was present on the protoarray, but was below our threshold and Kif2b was not present on the array. Nevertheless, all three proteins share high sequence similarity and we therefore tested whether these orthologs also interact with Fbxw. Indeed, Kif2b displayed a

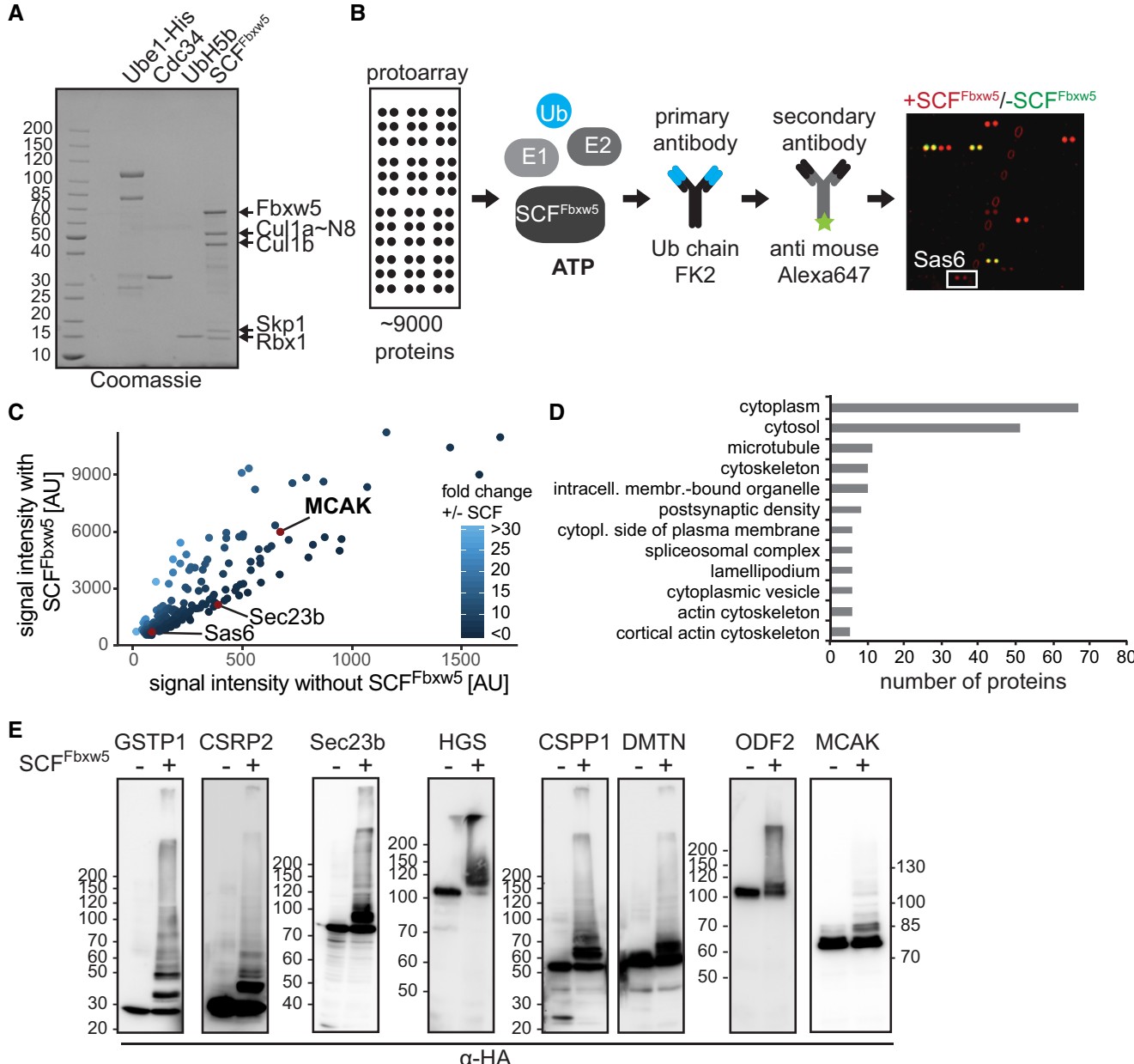

**Figure 1. Comprehensive substrate screening on protein microarrays (protoarray®) identifies 161 candidate substrates of SCF^Fbxw5.**

A  Coomassie-stained SDS–polyacrylamide gel electrophoresis (PAGE) of different proteins used in the assay. Cul1 was obtained using a split and co-express system (in which the C- and N-terminal domains are co-expressed as individual proteins and therefore run as two distinct bands at around 50 kDa) followed by *in vitro* neddylation. SCF^Fbxw5 complexes were prepared by mixing equimolar amounts of Fbxw5/Skp1 and Cul1~Nedd8/Rbx1 sub-complexes. Numbers left of the gel indicate molecular weight marker in kilo-Dalton (kDa, same accounts for all following gels and blots).

B  Workflow and example of a sub-array of the protoarray screen. Protein microarrays containing more than 9,000 human proteins spotted in duplicates were incubated with 15 µM FITC-labelled ubiquitin (Ub), 100 nM E1 (Uba1-His$_6$), E2s (0.5 µM each of UbcH5b and Cdc34) and 0.15 µM SCF^Fbxw5 for 1.5 h at 37°C. Right panel shows overlay of a selected sub-array probed with (red) or without (green) SCF^Fbxw5 complexes. White box marks the established substrate Sas6.

C  Comparison of protoarray signal intensities of candidate substrates probed with or without SCF^Fbxw5 complexes. Sas6, Sec23b and MCAK are marked as red dots (other published substrates (e.g. Ask1, Eps8) were not among the 9,000 proteins spotted on the array). Note that axes have different scaling.

D  Cellular components GO analysis of identified substrates using DAVID webtool with protoarray proteins as background.

E  Validation of individual targets by manually curated *in vitro* ubiquitylation experiments. HA-tagged (hemagglutinin) candidate proteins were purified from Hek293T cells via anti-HA immunoprecipitation (IP) followed by HA-peptide elution. Candidates were incubated with 20 µM His$_6$-Ubiquitin, 170 nM E1, E2s (0.5 µM each of UbcH5b and Cdc34) and 5 mM ATP in the presence or absence of 0.1 µM SCF^Fbxw5 for 2 h at 37°C. Substrates were detected via SDS–PAGE followed by Western blotting using anti-HA antibodies for detection.

Data information: Source data are presented in Dataset EV1.
Source data are available online for this figure.

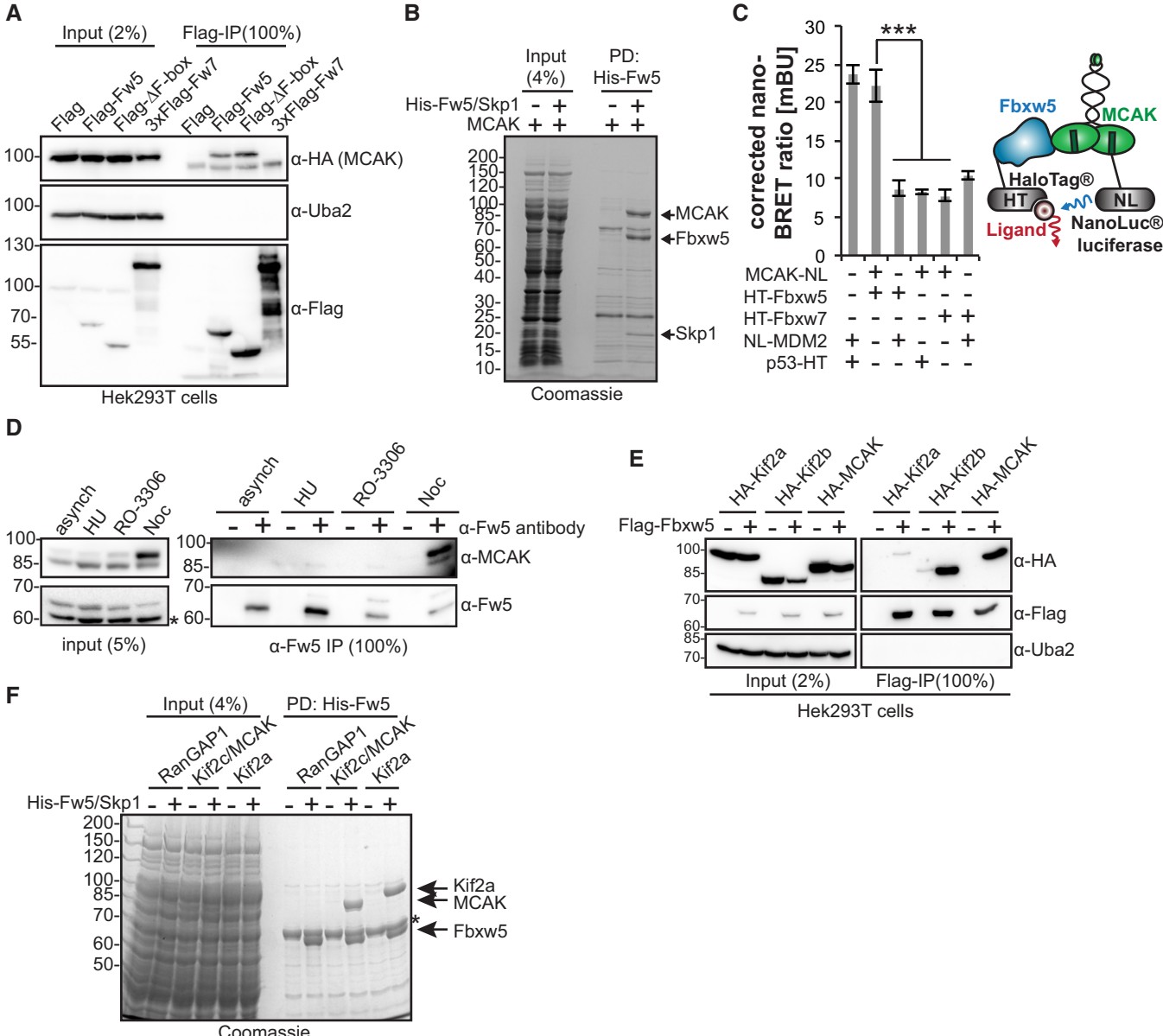

**Figure 2. Fbxw5 interacts with kinesin-13 family members.**

A  Co-IP of HA-MCAK with Flag-Fbxw5. Indicated proteins were expressed in Hek293T cells, extracted and subjected to anti-Flag IP followed by Western blot analysis.

B  *In vitro* pull-down experiment. Indicated proteins (purified from Sf21 cells, 5-fold molar excess of MCAK over Fbxw5/Skp1) were mixed with competing *Escherichia coli* lysate and precipitated via Ni-NTA agarose. Proteins were washed, eluted in SDS sample buffer and analysed by SDS–PAGE.

C  NanoBRET™ (Nano-bioluminescence resonance energy transfer) assay. Mdm2, p53 and Fbxw7 serve as controls. Mdm2 or MCAK were tagged with NanoLuc®-luciferase (Nluc), p53, Fbxw7 and Fbxw5 with HaloTag® (HT) and expressed in HeLa cells. After incubation with ligand overnight, substrate was added and plates were directly measured. Left: Quantification of four independent experiments. Error bars indicate standard deviation and asterisks the *P*-value of a two-tailed unpaired Student's *t*-test comparing each control with the MCAK/Fbxw5 pair (***$P < 0.001$). Right: Cartoon showing the NanoBRET™ principal.

D  Endogenous co-IP of MCAK with Fbxw5. RPE-1 cells were either grown asynchronously or arrested for 24 h with 2 mM HU (S phase), 10 μM RO-3306 ($G_2$) or 75 ng/ml nocodazole (M). Protein extracts were subjected to unspecific rabbit IgG (−) or anti-Fbxw5 (+) IP and analysed by Western blotting. Asterisk indicates an unspecific band detected by the Fbxw5 antibody.

E  Co-IP of HA-Kif2a, HA-Kif2b or HA-MCAK with Flag-Fbxw5 as in (A).

F  *In vitro* pull-down experiment as in (B). Asterisk indicates an unspecific protein from *E. coli*. Note: Kif2b purification from Sf21 cells was much less efficient, and it was therefore not included in the experiment.

Data information: Source data for (C) are presented in Source Data for Fig 2.
Source data are available online for this figure.

similar interaction with Fbxw5 in co-IPs and Kif2a precipitated weakly in these experiments (Fig 2E). However, it did bind efficiently to Fbxw5 in *in vitro* pull-down assays (Fig 2F). Taken together, our results demonstrate that Fbxw5 can directly recruit all three kinesin-13 proteins.

## SCF^Fbxw5 ubiquitylates MCAK in a highly efficient and specific manner

Considering that MCAK was a strong hit in our protoarray screen, we were surprised by its rather modest ubiquitylation efficiency within our validation experiments (Fig 1E). In order to investigate the ubiquitylation reaction in more detail, we generated neddylated SCF complexes in high amounts using the baculoviral expression system (Figs 3A and EV2A and B). One difference between the screen and the control experiments was the nature of ubiquitin (FITC-labelled vs His$_6$-tagged). We thus wondered whether the His$_6$-tag on ubiquitin had any impact on the ubiquitylation efficiency of MCAK. Indeed, while MCAK was only weakly ubiquitylated with His$_6$-tagged ubiquitin, the reaction became much more accelerated using untagged ubiquitin (Fig EV2C). Of note, ubiquitylation efficiency of Eps8 was not impaired by the His$_6$-tag (Fig EV2D). In addition, the effect was specific for the His$_6$-tag, as MCAK could be efficiently ubiquitylated with Flag-tagged ubiquitin (Fig EV2E). One explanation for this observation may be a higher positive net charge on the surface of MCAK that repels the His$_6$-tag on ubiquitin, but it shows in general that care must be taken when using tags in ubiquitylation experiments.

Using untagged ubiquitin for further characterisation, we could confirm that the slower migrating bands of MCAK in the reaction are due to modification by ubiquitin, as they were absent upon drop out of any essential component of the ubiquitin system (Fig 3B). Furthermore, we found that the reaction was much more pronounced using Cdc34 as an E2 compared to UbcH5b and combination of both did not further improve the reaction (Figs 3B and C, and EV2F). Unrelated proteins like Plk1 or RanGAP1 were not ubiquitylated (Fig 3D) and the reaction was highly specific to SCF^Fbxw5 as either replacing Cul1/Rbx1 with Cul4A/DDB1/Rbx1 (Figs 3D and EV2A, B and G) or Fbxw5 with other F-box proteins such as Fbxl2 or Fbxw7 (Figs 3E and EV2H and I) completely abolished MCAK ubiquitylation. Consistent with our previous findings on Eps8, *E. coli*-derived MCAK was also ubiquitylated (Fig EV2J), demonstrating that the reaction does not require preceding post-translational modifications on the substrate. In line with the observed binding capability, *E. coli*-derived Kif2a and Kif2b were also efficiently ubiquitylated (Fig 3F), showing that SCF^Fbxw5 is able to target all three orthologs.

Since ubiquitin can form chains of different linkage types with distinct physiological outcomes, we tested whether one of the two most prevalent ones (i.e. K48 and K63) is catalysed by SCF^Fbxw5 on MCAK. Using UbcH5b as an E2, the ubiquitylation pattern of MCAK was not affected by any of the mutants including methylated ubiquitin that lacks functional acceptor amino groups (Figs 3G and EV2K). This suggests that UbcH5b mainly catalyses mono-ubiquitylation on MCAK, as it has been shown also for the SCF^βTrCP substrate IκBα (Wu *et al*, 2010). Reactions using Cdc34 as E2 on the other hand showed massive loss of high molecular weight species of MCAK and Kif2a using either the K48R single or KK48,63RR double mutant, but not with K63R ubiquitin (Figs 3G and EV2K and L). The pattern of K48R was the same as for methylated ubiquitin and displayed high similarity with the one obtained with UbcH5b, demonstrating that SCF^Fbxw5 in concert with Cdc34 catalyses almost exclusively K48 chains on several lysine residues of MCAK and Kif2a. Consistent with this, mass spectrometry analysis of diGly-containing peptides after trypsin digestion of ubiquitylated MCAK revealed a total of 18 lysine residues that are modified by SCF^Fbxw5 and Cdc34 *in vitro* (Fig EV2M, Dataset EV2). Importantly, 15 of these 18 sites have been mapped previously in cell-based proteomic approaches (Udeshi *et al*, 2013; Hornbeck *et al*, 2015; Akimov *et al*, 2018) confirming the physiological relevance of our *in vitro* approach.

## SCF^Fbxw5 affects centrosomal MCAK levels in G$_0$

Since K48 chains provide an efficient signal for proteasomal degradation, we examined MCAK levels in RPE-1 cells upon Fbxw5 knockdown. Compared to non-targeting siRNA, MCAK amounts were slightly increased in asynchronously growing cells. Similar effects could be observed after G$_2$ arrest using the Cdk1 inhibitor RO-3306 (Vassilev *et al*, 2006) and upon mitotic arrest using nocodazole, in which MCAK levels were in general higher compared to the other treatments (Ganguly *et al*, 2008) (Fig 4A and B). However, this increase seems to be an effect of the prolonged mitotic arrest, as MCAK levels during normal mitosis were only moderately increased in thymidine-based synchronisation experiments (Fig EV3A). Interestingly, the most pronounced difference appeared in quiescent cells that had been serum-starved for 24 h. Here, MCAK levels went almost below detection in control samples, but were 4-fold higher upon Fbxw5 knockdown.

Due to this strong effect, we focused on serum-starved cells and used immunofluorescence (IF) to test whether a specific sub-population of MCAK becomes increased after Fbxw5 knockdown. We detected one prominent signal that co-localised with centrosomes (shown by ODF2 staining) and proved to be specific as it disappeared after MCAK knockdown (Fig 4C). In line with results from Western blotting, this signal was on average significantly stronger after Fbxw5 knockdown using three different siRNAs. As we could not observe an obvious Fbxw5 localisation at centrosomes (Fig EV3B and C), we wondered whether centrosomal MCAK represents a stable pool that would have to be degraded locally or whether it exchanges with cytoplasmic MCAK. To address this, we conducted fluorescence recovery after photobleaching (FRAP) experiments using a stable cell line expressing mNeonGreen-(mNG-) tagged MCAK under a doxycycline-inducible promoter. As shown in Fig EV3D, mNG-MCAK localisation was very similar to endogenous MCAK observed by IF. FRAP experiments of these cells upon doxycycline induction revealed that the centrosomal pool of mNG-MCAK is highly dynamic and recovers within the scale of seconds (Fig EV3E and Movie EV1), demonstrating that cytoplasmic Fbxw5 is perfectly capable to influence centrosomal MCAK levels.

## SCF^Fbxw5 and the APC/C regulate MCAK at distinct cell cycle stages

Considering the strong difference in MCAK levels upon serum starvation, we initially hypothesised that the Fbxw5-dependent regulation occurs upon mitotic exit when cells enter into a quiescent state.

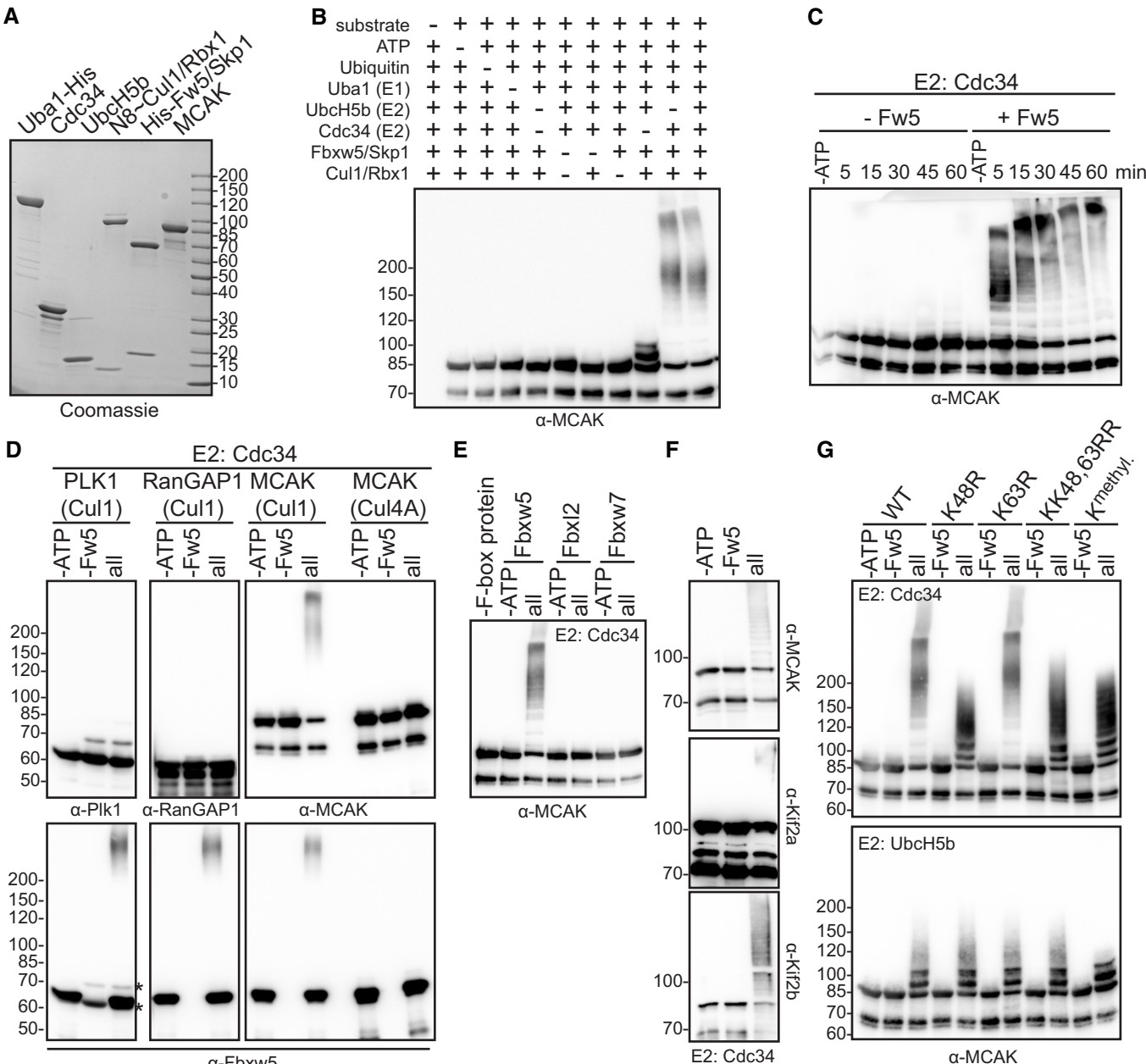

**Figure 3. SCF^Fbxw5 ubiquitylates MCAK in a highly efficient and specific manner.**

A  Coomassie-stained SDS–PAGE of different proteins used in the *in vitro* ubiquitylation assay. Cul1 was obtained as a full-length protein (and therefore runs at around 85 kDa in contrast to the split version of Fig 1) from Sf21 cells and *in vitro* neddylated (see Fig EV2A and B). SCF^Fbxw5 complexes for all ubiquitylation experiments were prepared by mixing equimolar amounts of Fbxw5/Skp1 and Cul1~Nedd8/Rbx1 sub-complexes.

B  Drop-out ubiquitylation experiment. 0.2 μM MCAK was incubated with 75 μM ubiquitin (untagged), 170 nM E1, 0.25 μM of UbcH5b and/or Cdc34 and 0.05 μM SCF^Fbxw5 for 15 min at 30°C followed by Western blotting using anti-MCAK antibodies for detection. Top labelling indicates component(s) that have been included (+) or omitted (−) from the reaction.

C  *In vitro* ubiquitylation experiment as in (B) using Cdc34 as E2, stopped at the indicated time points by adding SDS sample buffer.

D  *In vitro* ubiquitylation experiment as in (B). Blots 1&2: Plk1 or RanGAP1 used as substrates instead of MCAK. Blot 3: Left: Experiment as in (B). Right: Same reaction except that Cul1~Nedd8/Rbx1 complexes were replaced by Cul4A~Nedd8/DDB1/Rbx1 complexes (see Fig EV2A and B). Bottom blots: Same blots as above incubated with anti-Fbxw5 antibodies. Autoubiquitylation of Fbxw5 indicates activity of the E3 ligase complex. Asterisks indicate signals from Plk1 antibody (same species as anti-Fbxw5).

E  *In vitro* ubiquitylation experiment as in (B) except that different F-box/Skp1 sub-complexes (Fbxw5/Skp1, Fbxl2/Skp1 or Fbxw7/Skp1) were used (see Fig EV2H).

F  *In vitro* ubiquitylation experiment as in (B) using *Escherichia coli*-derived MCAK, Kif2a or Kif2b as substrates and Cdc34 as E2.

G  *In vitro* ubiquitylation experiment as in (B) using either wild type, K48R, K63R, KK48,63RR or methylated ubiquitin (K^methyl, see Fig EV2K). Top: Cdc34 used as E2. Bottom: UbcH5b used as E2.

Source data are available online for this figure.

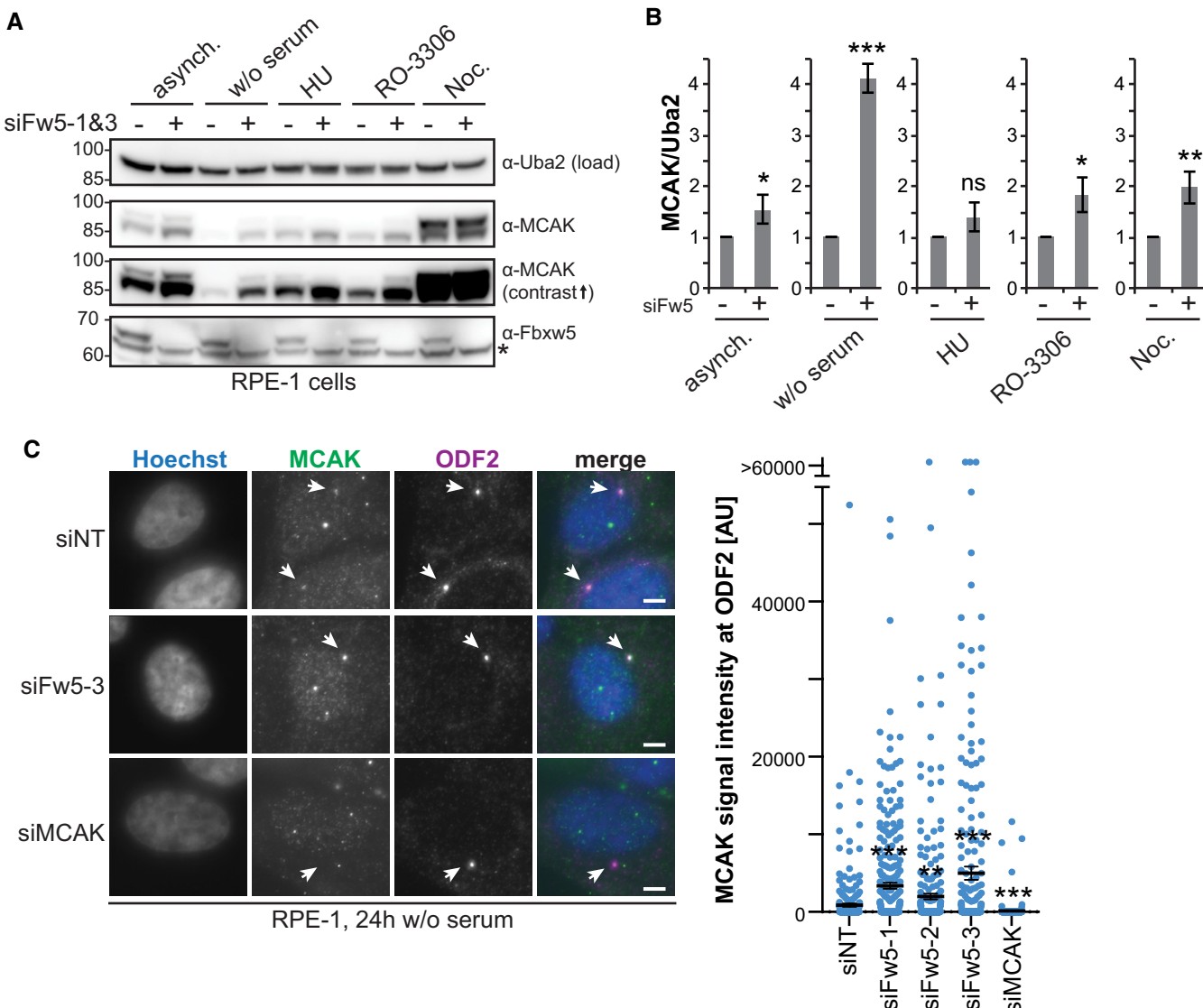

**Figure 4. SCF<sup>Fbxw5</sup> affects centrosomal MCAK levels in G₀.**

A Extracts of RPE-1 cells treated for 72 h with either non-targeting or Fbxw5-directed siRNAs (mix of siFw5-1&3) either grown asynchronously or arrested using 2 mM HU (S phase), 75 ng/ml nocodazole (M) and 10 μM RO-3306 (G₂) for 18 h or via serum starvation for 24 h (G₀). Indicated antibodies were used for detection. Asterisk indicates an unspecific band detected by the Fbxw5 antibody. Uba2 served as a loading control (same for all following blots).

B Quantification of the MCAK/Uba2 signal ratio normalised to each non-targeting control of four independent experiments. Error bars indicate standard deviation and asterisks the *P*-value of a two-tailed unpaired Student's *t*-test comparing each Fbxw5-directed siRNA sample with the according non-targeting control (*$P < 0.05$, **$P < 0.01$ and ***$P < 0.001$).

C RPE-1 cells were transfected with the indicated siRNAs (72 h total time), split on coverslips 24 h later and serum-starved for the last 24 h. Cells were fixed in methanol and analysed via immunofluorescence using the indicated antibodies together with Hoechst staining. Left: Maximum intensity projections (same for all following microscopy images) of representative images. Arrows indicate ODF2 signal. Scale bar = 5 μm. Note: Middle panel shows an example image of siFw5-3. Right: Quantification of MCAK signals co-localising with ODF2. Long black line shows mean intensity, and error bars indicate standard error of the mean of three independent experiments covering in total more than 280 cells. Asterisks indicate *P*-value of a Mann–Whitney test comparing each sample set with the non-targeting control (**$P < 0.01$ and ***$P < 0.001$).

Data information: Source data for (B) and (C) are presented in Source Data for Fig 4.
Source data are available online for this figure.

To test this, we used again an RPE-1 cell line expressing mNG-MCAK under a doxycycline-inducible promoter, transfected these cells with the according siRNAs and induced mNG-MCAK expression 24 h later. After another 24 h, we removed doxycycline and released cells into serum-free medium. Using spinning disk microscopy, we then selected cells in telophase and imaged them for 20 h with 20 min time intervals (Figs 5A and EV4A–C and Movie EV2). As expected, mNG-MCAK levels were generally increased upon

Fbxw5 knockdown at time point zero. To our surprise, mNG-MCAK signal gradually disappeared in both control and Fbxw5 knockdown cells with almost equal half-life times for the centrosomal population. Similar results were obtained for endogenous MCAK in an IF-based mitotic shake-off experiment, in which MCAK levels became gradually fainter at centrosomes in both control and knockdown cells (Fig EV4D).

The observation that endogenous and mNG-tagged MCAK slowly disappear upon entry into a quiescent $G_0$ state in an Fbxw5-

independent manner indicates the existence of another regulatory pathway. In line with this, proteasomal inhibition via MG-132 during the last 6 h of serum starvation significantly increased total MCAK levels in control but also in Fbxw5 knockdown cells (Fig 5B). One likely candidate for an alternative regulatory pathway is the anaphase-promoting complex/cyclosome (APC/C)—another multi-subunit ubiquitin E3 ligase that has been shown to target MCAK via Cdc20 in HeLa cells (Sanhaji *et al*, 2014) and via Cdh1 *in vitro* (Zhao *et al*, 2008; Singh *et al*, 2014). In order to test whether the

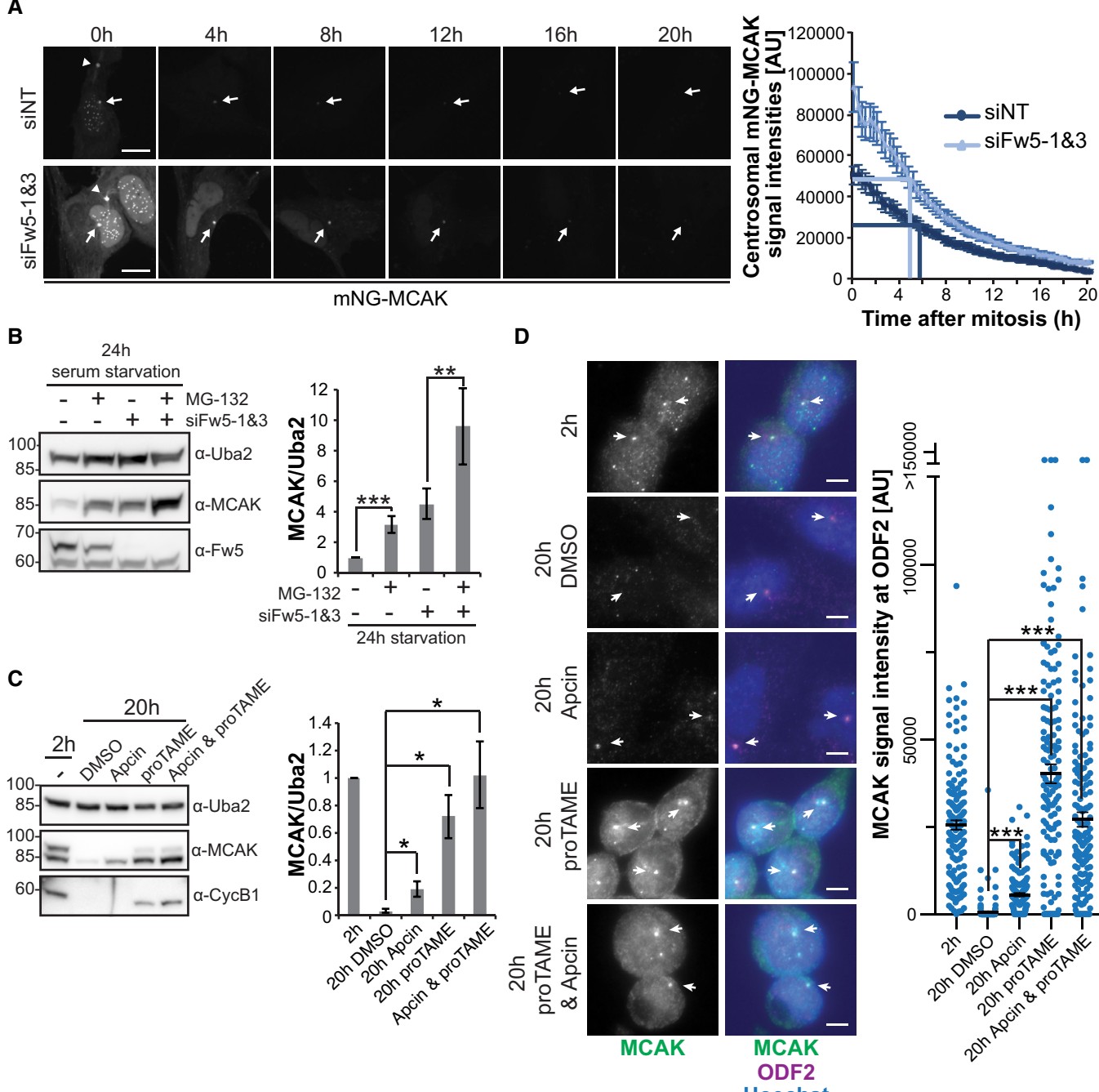

Figure 5.

**Figure 5. The APC/C regulates MCAK after mitosis.**

A Fluorescence-based pulse-chase experiment using a monoclonal RPE-1 cell line expressing mNG-MCAK under a doxycycline-inducible promoter. Cells were transfected with the indicated siRNAs (siFw5-1&3 together) and split 24 h later on Ibidi 8 Well Glass Bottom µ-slides while simultaneously inducing mNG-MCAK expression with 6 ng/ml doxycycline (pulse). 24 h later, doxycycline was washed out and cells were released into serum-free medium. Cells that have just finished or were about to finish mitosis (distinguishable for example by the midbody signal of mNG-MCAK [arrow heads]) were selected and imaged over 24 h, taking an image every 20 min (chase). Left: Representative images of selected time points. Scale bar = 10 µm. Arrows indicate centrosomal mNG-MCAK signals, arrowheads mNG-MCAK signals at the midbody. See also Movie EV2. Right: Quantification of centrosomal mNG-MCAK signals. Error bars show standard error of the mean of three independent experiments with 25 cells in total. *P*-value of a two-tailed unpaired Student's *t*-test comparing signal intensity of Fbxw5 knockdown over non-targeting control was mostly below 0.01 for time points 0–8 h and below 0.05 for time points 9–20 h. Dark and light blue lines indicate the time points at which mNG-MCAK signals are reduced by 50% compared to time point 0 for siNT and siFbxw5 samples, respectively.

B Extracts of RPE-1 cells treated with the indicated siRNA for 72 h, serum-starved for the last 24 h and treated either with DMSO (−) or 10 µM MG-132 (+) for the very last 6 h. Left: Representative blot. Right: Quantification of MCAK/Uba2 signal ratio normalised to non-targeting control with DMSO of five independent experiments. Error bars indicate standard deviation and asterisks the *P*-value of a two-tailed unpaired Student's *t*-test (\*\**P* < 0.01 and \*\*\**P* < 0.001).

C RPE-1 cells were treated with 75 ng/ml nocodazole for 4 h. Mitotic cells were shaken off, washed two times with phosphate-buffered saline (PBS) and released in serum-free medium. After 2 h, cells were either directly harvested (2-h sample) or treated with the indicated APC/C inhibitor for another 18 h before harvesting. Left: Representative blot. Right: Quantification of MCAK/Uba2 signal ratio normalised to the untreated, 2-h release sample. Error bars show standard deviation of three independent experiments and asterisks the *P*-value of a two-tailed unpaired Student's *t*-test (\**P* < 0.05).

D Same samples as in (C), except that cells grown on coverslips were fixed in methanol and analysed by immunofluorescence. Left: Representative images. Arrows indicate ODF2 signal. Scale bar = 5 µm. Right: Quantification. Error bars show standard error of the mean of three independent experiments, covering in total more than 150 cells. Asterisks indicate the *P*-value of a Mann–Whitney test (\*\*\**P* < 0.001).

Data information: Source data for (A, B, C, D) are presented in Source Data for Fig 5.
Source data are available online for this figure.

APC/C targets MCAK after mitotic exit, we again performed mitotic shake-off experiments with release into serum-free medium. This time, we added the APC/C inhibitors Apcin and proTAME after mitotic completion (~ 2 h after shake-off and nocodazole washout). While Apcin prevents substrate recognition by masking the D-Box binding site of Cdc20, proTAME inhibits incorporation of both substrate receptors (Cdc20 and Cdh1) into the APC/C complex. Combination of both compounds has been shown to completely block APC/C activity (Zeng *et al*, 2010; Sackton *et al*, 2014). As expected, total and centrosomal MCAK signals disappeared almost completely 20 h after mitosis in DMSO-treated samples (Fig 5C and D). However, this effect was partially reverted in Apcin-treated cells and almost completely reverted upon proTAME addition, suggesting that the APC/C is indeed responsible for MCAK removal upon entry into quiescence. Taken together, our findings show that after mitotic exit, MCAK is predominantly regulated by the APC/C.

## SCF$^{Fbxw5}$ regulates MCAK in G$_2$/M

The experiments described above did not reveal any stabilising effect on MCAK upon Fbxw5 depletion after mitotic exit. However, one striking difference that was revealed in the time course experiments (Figs 5A and EV4D) were the elevated levels of MCAK in Fbxw5 knockdown cells at time point zero. Accordingly, depletion of Fbxw5 led to elevated mNG-MCAK levels already in prometaphase (Fig EV4E and F). As mNG-MCAK expression was induced only for 24 h in this experiment, these results imply that Fbxw5 targets MCAK before mitotic exit. To address this, we employed cycloheximide (CHX) chase experiments in different pre-mitotic cell cycle arrests to further narrow down the exact timing of the Fbxw5-dependent regulation. Whereas MCAK remained relatively stable for 6 h in asynchronously growing and in S phase arrested cells, it became unstable in cells arrested in G$_2$ with the Cdk1 inhibitor RO-3306 in an Fbxw5-dependent manner (Fig 6A). Similarly, synchronising cells by double thymidine block revealed an Fbxw5-dependent destabilisation of MCAK within a 6 h release (Fig 6B),

which corresponds to an enrichment of cells in G$_2$ with intact Cdk1 activity (in contrast to the arrest with RO-3306). Together, these data are in line with our previous work on Eps8, which was also targeted during G$_2$ (Werner *et al*, 2013), indicating that SCF$^{Fbxw5}$ is particularly active during this cell cycle stage.

To gain further evidence that Fbxw5 targets MCAK specifically during G$_2$, we investigated MCAK amounts upon proteasomal inhibition. In contrast to asynchronously growing cells, MG-132 treatment under G$_2$ and M arrest led to a moderate but reproducible increase in MCAK amounts (Fig 6C). Since MG-132 treatment did not further increase MCAK levels upon Fbxw5 knockdown, these results confirm that MCAK becomes proteasomally degraded in an Fbxw5-dependent manner during G$_2$/M. Together, our results demonstrate that the activity of SCF$^{Fbxw5}$ towards MCAK starts during G$_2$ and becomes undetectable after mitotic exit.

## Fbxw5-dependent regulation of MCAK and its orthologs is required for ciliogenesis

Since excess activity of MCAK has been shown to increase mitotic duration (Maney *et al*, 1998; Bendre *et al*, 2016), we speculated that its regulation by Fbxw5 in G$_2$/M may be an essential process for mitosis. However, we did not observe a striking increase in the mitotic index of RPE-1 cells (Fig EV5A) and mild effects such as a delay in prometaphase may well be due to other Fbxw5 substrates like Sas6 or Eps8, which have been shown to impact on mitotic progression, too (Puklowski *et al*, 2011; Werner *et al*, 2013).

A second period during which elevated MCAK levels could be detrimental is the G$_0$ phase. In fact, overexpression of kinesin-13 proteins (including MCAK) has recently been demonstrated to impair ciliogenesis in RPE-1 cells upon serum starvation (Miyamoto *et al*, 2015). Consistent with this, we observed a mild but significant decrease in ciliated cells upon MCAK overexpression in serum-starved RPE-1 cells using our doxycycline-inducible system (Fig 7A and B). Moreover, cilia still present upon MCAK overexpression showed a significant reduction in their length that correlated with

MCAK intensities at basal bodies (Fig EV5B and C). Together, these results indicate that excess centrosomal MCAK negatively impacts on ciliogenesis.

Since loss of Fbxw5 increased MCAK levels during serum starvation (Fig 4C), we wondered whether Fbxw5 depletion induces similar defects in ciliogenesis as MCAK overexpression. Indeed,

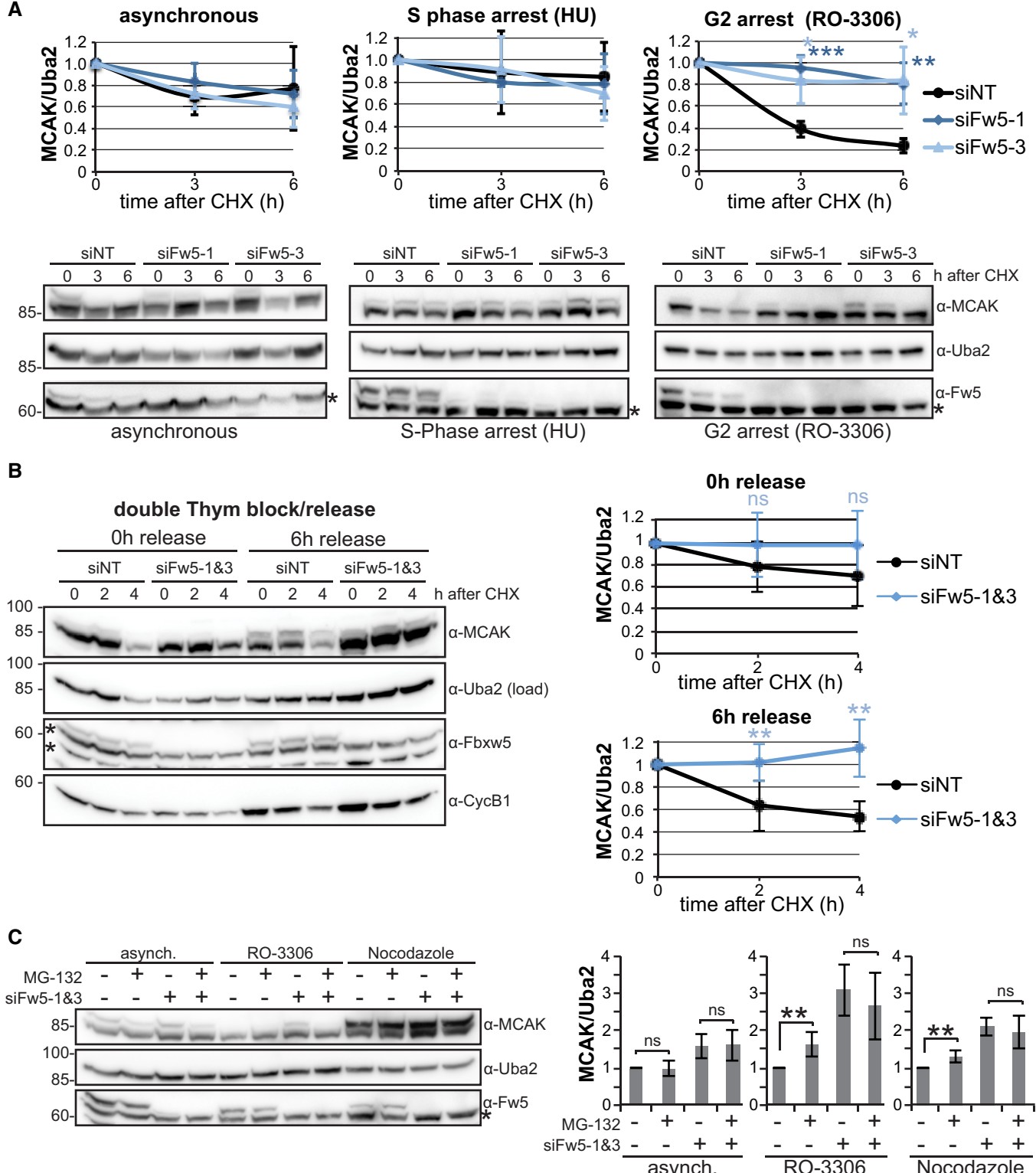

Figure 6.

**Figure 6.  $SCF^{Fbxw5}$ regulates MCAK in $G_2/M$.**

A   Cycloheximide (CHX) chase experiments of RPE-1 cells treated with the indicated siRNA for 48 h and then either grown further asynchronously or arrested in S phase with 2 mM hydroxyurea (HU) or in $G_2$ with 10 μM RO-3306 for 18 h. 50 μg/ml CHX was added to the cells, and samples were harvested at the indicated time points and analysed via SDS–PAGE and Western blotting using the indicated antibodies for detection. Bottom: Representative blots. Asterisks indicate an unspecific band detected by the Fbxw5 antibody. Top: Quantification of MCAK/Uba2 signal intensity ratios normalised to time point zero. Error bars show standard deviation of four independent experiments, and asterisks indicate $P$-value of a two-tailed unpaired Student's $t$-test (*$P < 0.05$, **$P < 0.01$ and ***$P < 0.001$).

B   CHX chase experiment as in (A), except that cells were treated with the indicated siRNAs for 48 h and arrested with 200 mM thymidine for 16 h, released for 9 h and again arrested for 16 h. Following the $2^{nd}$ release, 50 μg/ml CHX was added as indicated and samples were analysed as in (A). Left: representative blots. Asterisks indicate unspecific bands detected by the Fbxw5 antibody. Right: Quantification of MCAK/Uba2 signal intensity ratios normalised to time point zero. Error bars show standard deviation of six independent experiments, and asterisks indicate $P$-value of a two-tailed unpaired Student's $t$-test (**$P < 0.01$).

C   Extracts of RPE-1 cells treated with the indicated siRNAs for 72 h, grown asynchronously or arrested with the indicated compounds for the last 24 h and treated either with DMSO (−) or 10 μM MG-132 (+) for the very last 6 h. Left: Representative blots. Asterisk indicates an unspecific band detected by the Fbxw5 antibody. Right: Quantification of MCAK/Uba2 signal ratio normalised to non-targeting control with DMSO of 6 independent experiments. Error bars indicate standard deviation and asterisks the $P$-value of a two-tailed unpaired Student's $t$-test (**$P < 0.01$).

Data information: Source data for (A, B, C) are presented in Source Data for Fig 6.
Source data are available online for this figure.

knockdown of Fbxw5 with three different siRNAs led to a significant reduction in ciliated cells (Fig 7C and D) and remaining cilia under these conditions displayed on average much shorter axonemes (Fig EV5D). As we identified more than 150 candidate substrates of Fbxw5 in our screen, including many microtubule-associated proteins that could potentially influence ciliogenesis, we wondered whether this defect is at least partially due to stabilisation of MCAK and carried out rescue experiments. Strikingly, simultaneous knockdown of MCAK almost completely restored both number and length of cilia upon Fbxw5 depletion, suggesting that MCAK plays indeed an essential role within the observed phenotype (Fig 7C and D).

Since Kif2a and Kif2b have been described as negative regulators of ciliogenesis, too (Miyamoto *et al*, 2015), we tested also their involvement in the Fbxw5-dependent ciliogenesis defect. In line with our *in vitro* data, knockdown of Fbxw5 led to elevated levels of Kif2a and mNG-Kif2b upon serum starvation (Fig EV5E and F, note that we could not detect endogenous Kif2b with commercial antibodies and therefore had to overexpress the protein). Interestingly, simultaneous knockdown of Kif2a and Kif2b also rescued the Fbxw5-dependent ciliogenesis defect, albeit not as efficiently as MCAK depletion (Fig EV5G and H). In conclusion, our data

demonstrate that loss of Fbxw5 impairs ciliogenesis most likely due to elevated levels of kinesin-13 proteins.

# Discussion

Using comprehensive substrate screening on protein microarrays, we identified MCAK as a bona fide substrate of $SCF^{Fbxw5}$, assigned its regulation pathway to the $G_2/M$ phase of the cell cycle and demonstrated that this process is required for ciliogenesis upon entry into a quiescent state (Fig 7E). In addition, our *in vitro* ubiquitylation screening approach provides a useful and reliable source for potential Fbxw5 substrates. The number of 161 candidate substrates may seem high at first glance, but one has to keep in mind that proteins are probed here in a cell-free system. Fbxw5 may regulate some of its substrates only in certain cell types or during specific developmental stages. In contrast to cell-based screens, the protein microarray method is unique in yielding a comprehensive list of potential substrates without being limited to a particular cellular context. By providing a comprehensive substrate list, this approach could also be useful in identifying common motifs among targets. We tried to

**Figure 7.  Fbxw5 is required for ciliogenesis in an MCAK-dependent manner.**

A   Polyclonal RPE-1 cells expressing mNG-MCAK under a doxycycline-inducible promoter were seeded on coverslips, induced with 20 ng/ml doxycycline and grown for 24 h. Subsequently, cells were washed twice with PBS and incubated for another 24 h in serum-free medium without doxycycline. Cells were then fixed in methanol and subjected to IF using the indicated antibodies together with Hoechst staining. Left: Representative images. Scale bar = 10 μm. Right: Quantification of ciliated cells. Error bars show standard deviation of 5 independent experiments covering more than 200 cells, and asterisks indicate $P$-value of a two-tailed unpaired Student's $t$-test (**$P < 0.01$).

B   Immunoblot of extracts of cells used in (A) with the indicated antibodies for detection.

C   RPE-1 cells were treated with the indicated siRNA for 48 h followed by serum starvation for another 24 h. Cells were then fixed in methanol and subjected to IF using the indicated antibodies together with Hoechst staining. Left: Representative images. Note: Left panel shows an example image of siFw5-1. Scale bar = 10 μm. Right: Quantification of ciliated cells. Error bars show standard deviation of four independent experiments covering more than 200 cells, and asterisks indicate $P$-value of a two-tailed unpaired Student's $t$-test (*$P < 0.05$, **$P < 0.01$ and ***$P < 0.001$).

D   Immunoblot of RPE-1 cell extracts. Left: Representative blots. Asterisk indicates an unspecific band detected by the Fbxw5 antibody. Right: Quantification of MCAK/Uba2 signal ratio. Error bars show standard deviation of 6 independent experiments and asterisks the $P$-value of a two-tailed unpaired Student's $t$-test comparing each knockdown with the non-targeting control (**$P < 0.01$ and ***$P < 0.001$).

E   Model: MCAK is regulated by two distinct E3 ligases at different time points. While $SCF^{Fbxw5}$ targets MCAK for degradation during $G_2/M$, the APC/C takes over after mitotic exit. Via the $SCF^{Fbxw5}$ pathway, cells ensure that levels of MCAK and its orthologs are kept low upon entry into $G_0$ and thus permit ciliogenesis in the following cell cycle.

Data information: Source data for (A, C, D) are presented in Source Data for Fig 7.
Source data are available online for this figure.

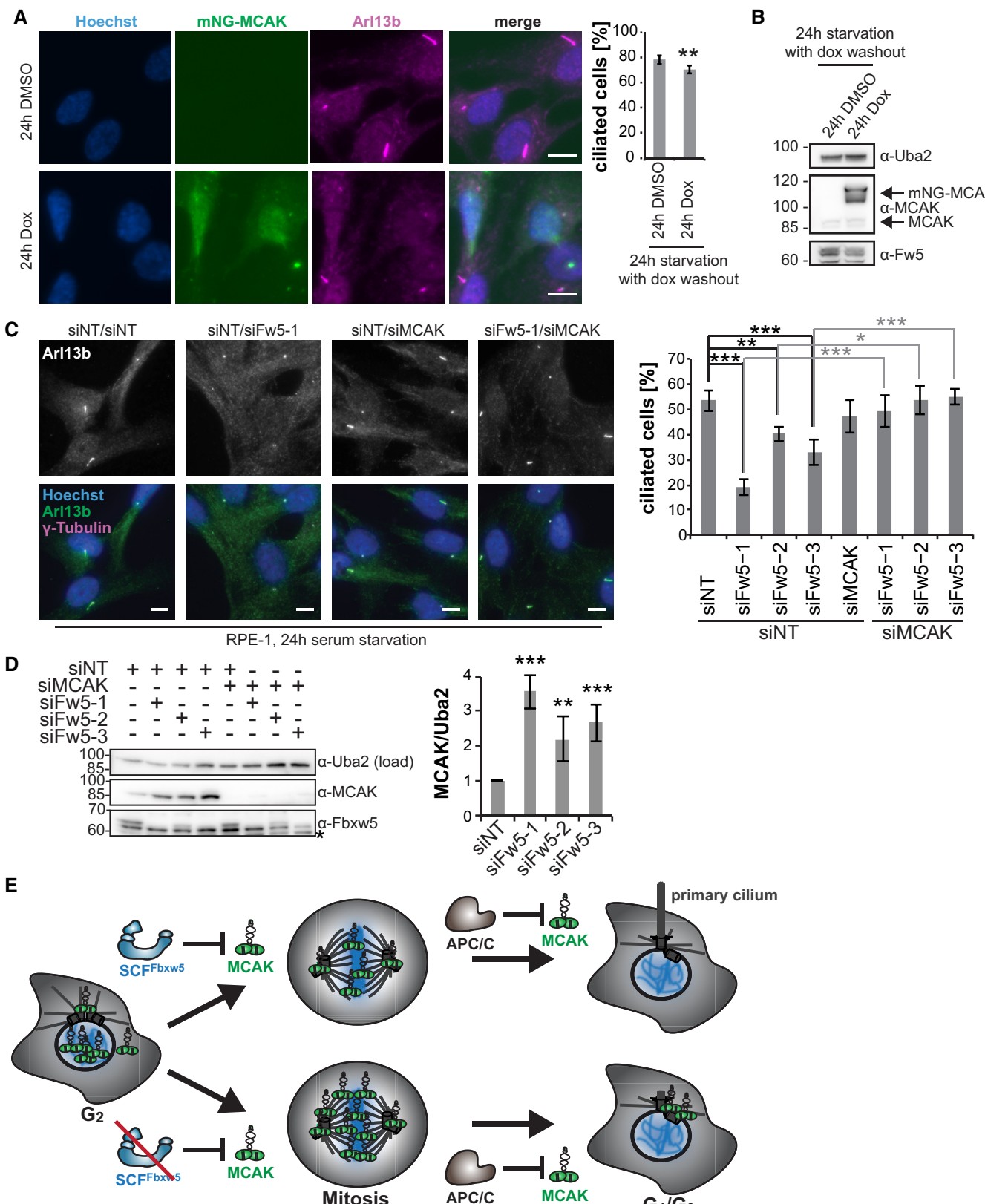

**Figure 7.**

identify an Fbxw5 degron using different motif search algorithms (DILIMOT, STREME, MEME, GLAM2) (Neduva et al, 2005; Frith et al, 2008; Bailey et al, 2009; Bailey, 2021), but were unable to detect an enriched motif (including the one identified for Sec23b (Jeong et al, 2018)), which suggests that substrate recruitment of Fbxw5 may be more variable than observed for other F-box proteins. Structural analysis of Fbxw5 in complex with different substrates would certainly help to define such a low complexity degron.

The identification of MCAK and its orthologs as substrates of SCF$^{Fbxw5}$ is a critical step towards a better understanding of these important microtubule depolymerases. Besides their well-explored roles in mitosis, MCAK and its orthologs have been identified as negative regulators of ciliogenesis (Miyamoto et al, 2015). Here, we show that all three proteins, MCAK, Kif2a and Kif2b, are ubiquitylated in vitro by SCF$^{Fbxw5}$ and become stabilised upon Fbxw5 knockdown in serum-starved cells. Importantly, knockdown of MCAK, Kif2a or Kif2b could restore cilia formation, suggesting that all three proteins contribute to the defect, possibly even in a synergistic manner. This could explain why overexpression of MCAK alone had only mild effects on ciliogenesis, compared to the much stronger defect upon Fbxw5 depletion. Other substrates of Fbxw5 may also contribute to the phenotype, but the fact that knockdown of MCAK or its orthologs almost completely restored ciliogenesis argues that it is based predominantly on increased kinesin-13 activity.

How MCAK regulates ciliogenesis is still elusive. Besides a reduction of ciliated cells, we also observed a shortening of remaining cilia in Fbxw5 knockdown or MCAK overexpressing cells. Taking into account its ability to depolymerise microtubules at both ends (Hunter et al, 2003), MCAK could indeed exert its microtubule activity at axonemes, similarly to what has been proposed for Kif2a (Miyamoto et al, 2015). Newly synthesised axonemal microtubules may be better accessible for MCAK-dependent depolymerisation due to their shorter length, a potentially incomplete microtubule lattice or a lack of post-translational modifications that could impair MCAK activity such as detyrosination (Peris et al, 2009). Whereas further work is certainly necessary to determine the exact mode of action, this theory could also explain why Fbxw5 targets MCAK in $G_2/M$. Such a preceding regulatory event may be required to guarantee sufficient reduction of MCAK levels at the beginning of the quiescent state in order to allow timely formation of primary cilia. Our time-lapse analysis indicates that the APC/C-dependent degradation of MCAK after mitosis is rather slow, with a half-life of about 6 h (Fig 5A). Thus, cells deficient of Fbxw5 would reach MCAK levels of unperturbed cells only after 6 h, which depicts a massive delay considering that RPE-1 cells form cilia already after 8 h in serum starvation (Kurtulmus et al, 2018). Taking into account the concomitant increase in the amounts of Kif2a and Kif2b, this can explain why the APC/C is apparently unable to compensate for loss of Fbxw5. Such a delay could be particularly detrimental in settings that require fast cilia formation, for example during tissue and organ development. Hence, placing the regulation process towards the end of the preceding cell cycle may ensure timely removal of negative factors enabling quick cilia-dependent sensing of extracellular signals later on.

Could this be a general concept? To the best of our knowledge, such a preceding regulatory process for ciliogenesis has not been described so far. However, degradation of the centriolar protein CP110 by SCF$^{CyclinF}$ has been shown to occur also during $G_2/M$, where it ensures centrosome homeostasis during mitosis (D'Angiolella et al, 2010; Li et al, 2013). Other studies have established that CP110 also counteracts ciliogenesis by recruiting kinesin-13 Kif24 (Spektor et al, 2007; Kobayashi et al, 2011). As far as we know, it has not been demonstrated yet whether the SCF$^{CyclinF}$-dependent regulation event is indeed required for ciliogenesis. But if this turns out to be the case, degradation of negative factors by SCF E3 ligases during $G_2/M$ may represent a common mechanism that allows timely formation of primary cilia in the following $G_1/G_0$ phase. It will be interesting to see how such preceding regulatory events impact on cilia-dependent developmental programmes within multicellular organisms and whether they play a critical role in ciliopathies.

# Materials and Methods

### Cell culture

RPE-1 hTERT cells (ATCC: CRL-4000) were cultured in Dulbecco's modified Eagle medium/Nutrient mixture F12 (DMEM/F12) supplemented with 10% filtrated foetal bovine serum (FBS). Hek293T cells (DSMZ: ACC 635 Lot 4) were cultured in DMEM supplemented with 2 mM glutamine and 10% FBS. Hela cells were cultured in DMEM Glutamax™ supplemented with 10% FBS. All cell lines were authenticated at the DKFZ Genomics and Proteomics Core Facility (Heidelberg), regularly tested for mycoplasm contamination, kept constantly below confluency and were cultivated at 37°C with 5% $CO_2$ for no longer than 8 weeks (~ 20 passages). In order to induce ciliogenesis, cells were washed 2× with phosphate-buffered saline (PBS) and then incubated with DMEM/F12 medium lacking FBS.

### Plasmid transfection

Hek293T cells were transiently transfected 1 day after seeding in 15-cm dishes at a confluency of 50% with 30 µg total plasmid DNA using polyethyleneimine (PEI, 1 mg/ml pH 7.0) at a DNA:PEI ratio of 1:2.5 in serum-free DMEM. Serum supplemented DMEM was added 6 h after transfection. Cells were harvested 24–48 h after transfection.

### siRNA transfection

siRNA transfection was performed with Lipofectamine RNAi MAX (Invitrogen) according to the manufacturer's recommendation for reverse transfection. For a 6-well format, trypsinised cells were mixed with 500 µl Opti-MEM (Gibco) containing 4 µl RNAi MAX and 50 pmol total siRNA and medium was added to reach a total volume of 2.5 ml. After ~ 6 h, cells were washed 1x with PBS, replenished with fresh medium and split on the next day. If not otherwise mentioned, cells were harvested or analysed 48 or 72 h after siRNA transfection. siRNAs and all other consumables are listed in Appendix Table S2.

### Generation of stable cell lines

For the generation of stable cell lines expressing mNG-MCAK, mNG-Fbxw5 and mNG-Kif2b under a doxycycline-inducible promoter, the retroviral transduction system of Clonetech® "Retro-X™ Tet-On®

3G Inducible Expression System" was used according to the manufacturer's protocol (retroviral gene transfer and expression user manual). Stable cells expressing low levels of the according protein were sorted using a BD FACSMelody™ or BD FACSAria III (Becton Dickinson) after induction with doxycycline for 24 h.

## Insect cell Culture

Sf21 cells were cultured in Sf-900™ II SFM or ExCell® 420 Serum-Free Medium at 130 rpm 27°C and generally kept between $1 \times 10^6$ and $8 \times 10^6$ cells/ml. Baculoviruses for protein expression were generated using the Bac-to-Bac Baculovirus Expression System (Invitrogen) according to the manufacturer's instructions. For bacmid generation, the according plasmid was transformed into DH10MultiBac chemically competent cells. Bacmids were isolated, tested for integrity by PCR and transfected into Sf21 cells using Cellfectin-II reagent in a 6-well format. 72 or 96 h after transfection, medium of transfected cells was harvested and filtered to generate P1 viral stocks. High-titre P2 stocks were produced by infecting 50 ml of Sf21 cells with the according P1 stock for 72 h and used for final protein expression by infecting 100–200 ml of Sf21 cells at $2 \times 10^6$ cells/ml for 48 h. All F-box proteins were purified together with Skp1 by co-infecting Sf21 cells with both P2 viral stocks.

## Protoarray *in vitro* ubiquitylation screen

Human ProtoArray® Microarray (v5.0) containing 9,483 unique human proteins expressed as GST-fusion in insect cells was purchased from Invitrogen. Arrays were blocked in 4°C cold blocking buffer (50 mM HEPES pH 7.4, 200 mM NaCl, 0.05% Tween-20, 25% glycerol, 10 mg/ml ovalbumin, 5 mM reduced glutathione, 1 mM dithiothreitol (DTT), 1 µg/ml aprotinin/pepstatin/leupeptin) for 1.5 h. Arrays were then washed with assay buffer (50 mM Tris pH 7.5, 100 mM NaCl, 5 mM MgCl₂, 0.05% Tween-20, 2 mg/ml ovalbumin, 1 mM DTT, 1 µg/ml aprotinin/pepstatin/leupeptin) for 5 min. For the reaction, 120 µl ubiquitylation reaction mixture (100 nM Uba1, 0.5 µM UbcH5b, 0.5 µM Cdc34, 15 µM FITC-Ubiquitin, 150 nM SCF[Fbxw5] (containing split-Cul1 from *E. coli* (Li *et al*, 2005)), energy regeneration solution (B-10, Boston Biochem) in assay buffer were carefully pipetted directly on the arrays. The slides were covered with cover slips and incubated in a humid chamber for 1.5 h at 37°C. Reactions were stopped by submerging slides in 5 ml assay buffer and removing the cover slip. Assay buffer was aspirated immediately and slides were washed twice with BSA buffer (PBS (140 mM NaCl, 2.7 mM KCl, 10 mM Na₂HPO₄, 1.5 mM KH₂PO₄, pH 7.5), 0.1% Tween-20, 1% BSA). Slides were then washed 3 times with 0.5% SDS followed by three more washes with BSA buffer. For detection of ubiquitylated proteins, slides were first blocked with antibody blocking buffer (PBS, 0.1% Synthetic Block [Invitrogen]) for 10 min followed by 1.5 h incubation at 4°C with anti-ubiquitin chain antibody FK2 (1 µg/ml FK2 in antibody blocking buffer). Modified proteins were finally labelled with Alexa 647-labelled anti-mouse antibodies. Slides were washed, spin-dried (200 *g* for 1 min) and scanned with a GenePix 4000B microarray scanner (Molecular Devices, made available by EMBL Genomics Core Facility). GenePix® Pro v.6 software (Molecular Devices) was used to align protein spots and quantify fluorescence intensity of the spots. Background correction was performed with Protein

Prospector analyzer (Invitrogen) followed by normalisation using protein array analyzer package (v1.3.3) (Turewicz *et al*, 2016) for R (Version 3.2.1.). Proteins were considered SCF[Fbxw5]-dependent ubiquitylation targets if signal intensity was > 500 and more than 5-fold increased over arrays incubated without E3 ligases. All candidate spots were manually controlled on the scanned images.

## GO analysis and statistical analysis

GO analysis of cellular components (CC_Direct) was performed using the DAVID Bioinformatics Resources 6.8. SwissProt IDs of candidate proteins were converted to EntrezGene ID before analysis via http://www.uniprot.org. SwissProt IDs of all proteins in the Human ProtoArray® Microarray (v5.0) were converted to Entrez-Gene ID accordingly and used as background.

All statistical analyses were performed using either Microsoft Excel® (for Student's *t*-test) for Mac (version 14.6.8) or GraphPad Prism v9 for Mac (for Mann–Whitney test and Pearson correlation). For Western blot quantification, ratio of MCAK signal over Uba2 signal was calculated and normalised to the non-targeting control and statistical significance was assessed using a two-tailed unpaired Student's *t*-test. Due to thresholding, MCAK signal analysis using immunofluorescence yielded many values of zero intensity and thus generated a non-Gaussian distribution. Hence, a Mann–Whitney test was used to estimate statistical significance.

## Recombinant protein purification

All bacteria-derived proteins were expressed in Rosetta (DE3) cells, all insect cell-derived proteins in Sf21 cells. After purification, final protein samples were aliquoted, snap-frozen in liquid $N_2$ and stored at −80°C until further use. Expression and purification of RanGAP1, Eps8, Ubiquitin, Uba1, UbcH5b, UbcH5c, Cdc34, APPBP1-Uba3, Nedd8, Ubc12 and Nedd8*Cul1A-Cul1B/Rbx1 was described before (Mahajan *et al*, 1997; Disanza *et al*, 2004; Chiba, 2005; Huang & Schulman, 2005; Pickart & Raasi, 2005; Werner *et al*, 2013). For Ube2G1 expression, the same procedure as for UbcH5b and Cdc34 was followed. Briefly, proteins were expressed overnight in Rosetta (DE3) cells at 16°C with 200 µM IPTG, lysed in 50 mM Tris pH 8, 300 mM NaCl, 1 mM EDTA, 1 mM β-mercaptoethanol, 1 µg/ml aprotinin/pepstatin/leupeptin by passing two times through an Emulsion Flex-C5 microfluidizer. Afterwards, glutathione-Sepharose beads were added and extensively washed and proteins were eluted in 50 mM Tris pH 8.8, 150 mM NaCl, 1 mM β-mercaptoethanol and 25 mM glutathione. Finally, GST-tag was cleaved with PreScission protease overnight and removed by molecular sieving using an analytical Superdex 75 10/300 chromatography column (GE Healthcare).

For His-tagged MCAK and Kif2a purified from *E. coli*, protein expression was induced overnight at 16°C with 200 µM IPTG and cells were lysed by passing two times through an Emulsion Flex-C5 microfluidizer in buffer A (25 mM Hepes pH 7.4, 300 mM KCl, 2 mM MgCl₂, 5 mM β-mercaptoethanol, 30 mM imidazole, 50 µM ATP, 1 µg/ml aprotinin/pepstatin/leupeptin and 1 mM PefaBloc [Roche]). Proteins were purified over Ni-NTA, eluted in buffer A (devoid of ATP and PefaBloc but containing 200 mM instead of 30 mM imidazole) and further purified over a Superdex 200 10/300 GL column (GE Healthcare) in buffer B (25 mM Hepes, pH 7.4,

300 mM KCl, 2 mM MgCl$_2$, 1 mM DTT, 1 µg/ml aprotinin/pepstatin/leupeptin). His-tagged Kif2b was generated accordingly, except that lysis was performed in buffer C (50 mM sodium phosphate buffer pH 6.0, 300 mM KCl, 2 mM MgCl$_2$, 5% glycerol, 5 mM β-mercaptoethanol, 30 mM imidazole, 0.06% NP-40, 50 µM ATP, 1 µg/ml aprotinin/pepstatin/leupeptin and 1 mM PefaBloc). Proteins were eluted in buffer C devoid of ATP, NP-40 and PefaBloc but containing 200 mM imidazole and further purified over a Superdex 200 10/300 GL column in buffer B.

For MCAK and Kif2a purified from insect cells, 200 ml of Sf21 cells at $2 \times 10^6$ cells/ml were infected with 1% (v/v) of the according baculovirus P2 stock for 48 h. Cells were lysed in buffer A by sonication ($3 \times 5$ s on, 20 s off at 40% amplitude), and proteins were purified via Ni-NTA and eluted in buffer A devoid of ATP, aprotinin/pepstatin/leupeptin and PefaBloc but containing 200 mM imidazole. Next, His-tag was removed by overnight TEV cleavage with simultaneous dialysis against buffer A at 4°C. Afterwards, excess His-tagged TEV was removed with Ni-NTA and proteins were further purified over a Superdex 200 10/300 GL column in buffer B.

For expression of Fbxw5, Fbxl2 and Fbxw7, 200 ml of Sf21 cells at $2 \times 10^6$ cells/ml were simultaneously infected with each 1% (v/v) F-box protein- and Skp1-expressing baculovirus P2 stocks for 48 h followed by lysis in buffer D (20 mM Na-phosphate buffer pH 8.0, 300 mM NaCl, 10 mM imidazole, 1 mM β-mercaptoethanol, 5% glycerol, 1 µg/ml aprotinin/pepstatin/leupeptin and 1 mM Pefabloc) by sonication ($3 \times 5$ s on, 20 s off at 40% amplitude). Proteins were purified via Ni-NTA, eluted in buffer D devoid of PefaBloc but containing 200 mM imidazole and further purified over a Superdex 200 10/300 GL column in Buffer E (20 mM HEPES pH 7.3, 110 mM K-acetate, 2 mM Mg-acetate, 1 mM EGTA, 1 µg/ml aprotinin/pepstatin/leupeptin).

### Generation of neddylated Cul1 and Cul4A complexes

The bacteria-derived split Cul1A-Cul1B/Rbx1 sub-complex was obtained from B. Schulman. Procedures for generation of neddylated Cul1/Rbx1 and Cul4A/DDB1/Rbx1 from insect cells were modified from Fischer *et al* (2011), Li *et al* (2005) and Scott *et al* (2016). For Cul1/Rbx1, 200 ml of Sf21 cells at $2 \times 10^6$ cells/ml were infected with 1% (v/v) of the baculovirus P2 stock for 48 h. Cells were lysed in buffer F (50 mM Tris pH 8, 200 mM NaCl, 5 mM DTT and 30 mM imidazole) by passing the cell suspension twice through an Emulsion Flex-C5 microfluidizer. Proteins were first purified via Ni-NTA, eluted in buffer F containing 200 mM imidazole and further purified over a 1 ml Resource S cation exchange chromatography column (GE Healthcare) using a 20 ml gradient from 100 to 500 mM NaCl in 50 mM 2-(*N*-morpholino)ethanesulphonic acid (MES) pH 6.5 and 5 mM DTT. Major peak fractions eluting at around 200 mM NaCl were pooled, concentrated and further purified over a Superdex 200 10/300 GL column in buffer G (50 mM Hepes pH 7.4, 200 mM NaCl, 2 mM DTT). His-Cul1/Rbx1 containing fractions were pooled, concentrated and neddylated at 8 µM with 0.1 µM APPBP1/Uba3, 1 µM Ubc12 and 1 mM ATP in neddylation buffer (25 mM Hepes pH 7.4, 200 mM NaCl and 10 mM MgCl$_2$). Reaction was started by addition of 20 µM Nedd8, incubated at 30°C for 5 min and stopped by the addition of 10 mM DTT on ice. Components of the neddylation machinery were then

removed by another round of Ni-NTA purification. Eluted Nedd8˜His-Cul1/Rbx1 was again concentrated, TEV cleaved overnight at 4°C and further purified over Superdex 200 10/300 GL column in buffer G. Complete neddylation was confirmed via SDS–PAGE (Fig EV2B).

Neddylated Cul4A/DDB1/Rbx1 complexes were generated essentially in the same way, except that the cation exchange step was replaced by an anion exchange procedure using a 1 ml MonoQ 5/50 anion exchange chromatography column over a 20 ml gradient ranging from 150 to 500 mM NaCl in 50 mM Tris pH 8 and 5 mM DTT. Here, Cul4A/DDB1/Rbx1 eluted at around 250 mM NaCl. All following steps including the neddylation reaction and TEV cleavage were conducted as for Cul1.

### Purification of HA-tagged proteins from Hek293T cells

For purification of HA-tagged proteins for *in vitro* ubiquitylation experiments, transiently transfected Hek293T cells from $4 \times 15$ cm dishes were collected 48 h after transfection, lysed in RIPA buffer (50 mM Tris pH 8.0, 300 mM NaCl, 1% NP-40, 0.5% Na-deoxycholate, 0.1% SDS, 1 µg/ml aprotinin/pepstatin/leupeptin, 1 mM PefaBloc), cleared by high-speed centrifugation and incubated for 3 h with 10 µl anti-HA-agarose. Next, beads were washed twice with wash buffer (20 mM HEPES pH 7.3, 110 mM K-acetate, 2 mM Mg-acetate, 1 mM EGTA, 0.1% NP-40, 1 µg/ml aprotinin/pepstatin/leupeptin, 20 mM *N*-ethylmaleimide [NEM]). Afterwards, beads were washed two more times with wash buffer devoid of NEM. Proteins were eluted by three consecutive elutions at 25°C for 10 min with one bead volume of elution buffer (20 mM HEPES pH 7.3, 100 mM NaCl, 110 mM K-acetate, 2 mM Mg-acetate, 1 mM EGTA, 0.2 mg/ml ovalbumin, 0.1% NP-40, 1 µg/ml aprotinin/pepstatin/leupeptin, 0.2 mg/ml HA peptide and 50 µM PR619) under constant agitation. Eluates were combined, aliquoted, snap-frozen and stored at −80°C until further use.

### *In vitro* ubiquitylation

*In vitro* ubiquitylation was performed as described previously (Werner *et al*, 2013) with minor alterations. SCF complexes were generated by mixing equimolar amounts of the F-box protein/Skp1 sub-complex with either Cul1/Rbx1 or Cul4A/DDB1/Rbx1 sub-complexes. Ubiquitylation mixes were prepared on ice in a total volume of 20 µl. The reaction was started by adding ATP and stopped by adding 20 µl 2× sample buffer (100 mM Tris pH 6.8, 2% (w/v) SDS, 0.2% (w/v) bromophenol blue, 20% glycerol, 200 mM DTT). For proteins purified from Hek293T cells (Fig 1E), 1–5 µl of HA-tagged candidate substrates were incubated with 170 nM Uba1, 0.5 µM UbcH5b, 0.5 µM Cdc34, 20 µM His$_6$-Ubiquitin, 5 mM ATP in the presence or absence of 100 nM SCF$^{Fbxw5}$ in SAB$^+$ buffer (20 mM HEPES pH 7.3, 110 mM K-acetate, 2 mM Mg-acetate, 1 mM EGTA, 0.2 mg/ml ovalbumin, 0.05% Tween-20, 1 µg/ml aprotinin/pepstatin/leupeptin) at 37°C for 2 h. If not otherwise mentioned, substrates purified from bacteria or insect cells were used at 0.2 µM and in general incubated with 170 nM Uba1, 0.25 µM E2, 75 µM Ubiquitin and 5 mM ATP in the presence or absence of 50 nM SCF$^{Fbxw5}$ in SAB+ buffer. The bacteria-derived split Cul1A-Cul1B/Rbx1 sub-complex obtained from B. Schulman was used for Fig 1, the insect cell-derived Cul1/Rbx1 sub-complex for all other

reactions. If not otherwise specified, Sf21-derived MCAK was used as substrate.

## Mass spectrometry

### In-gel digestion

Protein bands of interest were manually excised from gels. The gel pieces were washed once with 60 μl of 1:1 (v/v) 50 mM triethylammonium bicarbonate buffer (TEAB; Sigma-Aldrich, Taufkirchen, Germany) and acetonitrile (ACN; Roth, Karlsruhe, Germany), pH 8.5 for 10 min and shrunk three times for 10 min each in 60 μl ACN and washed in 60 μl 50 mM TEAB, pH 8.5.

Gel pieces were dehydrated with 60 μl 100% ACN. A total of 70 μl of 8 ng/μl in 50 mM TEAB trypsin solution (sequencing grade, Thermo Fisher, Rockford, USA) was added to the dry gel pieces and incubated 4 h at 37°C. The reaction was quenched by addition of 20 μl of 0.1% trifluoroacetic acid (TFA; Biosolve, Valkenswaard, The Netherlands). The resulting peptides were extracted once for 15 min with 50 μl 1:1 (v/v) 50 mM TEAB and ACN, pH 8.5 and once for 15 min in 70 μl ACN. The supernatant from each extraction step was collected and dried in a vacuum concentrator diluted in 15 μl 0.1% TFA. 5 μl were injected into LC.

### LC-MS measurements

Nanoflow LC-MS$^2$ analysis was performed with an Ultimate 3000 liquid chromatography system coupled to an Orbitrap Q Exactive mass spectrometer (Thermo Fisher, Bremen, Germany). Samples were delivered to an in-house packed analytical column (inner diameter 75 μm × 20 cm; CS—Chromatographie Service GmbH, Langerwehe, Germany) filled with 1.9 μm ReproSil-Pur-AQ 120 C18 material (Dr. Maisch, Ammerbuch-Entringen, Germany). Solvent A was 0.1% formic acid (FA; ProteoChem, Denver, CO, USA) in $H_2O$ (Biosolve), and solvent B was composed of 0.1% FA (ProteoChem), 10% $H_2O$ (Biosolve) and 89.9% ACN (Biosolve). Sample was loaded to the analytical column for 20 min with 3% B at 550 nl/min flow rate. Peptides were separated with 25 min linear gradient (3–40% B) with flow rate of 300 nl/min.

The Q Exactive mass spectrometer was operated in data-dependent acquisition mode, automatically switching between MS, acquired at 60,000 ($m/z$ 400) resolution, and MS$^2$ spectra, generated for up to 15 precursors with normalised collision energy of 27% in the HCD cell and measured in the Orbitrap at 15,000 resolution. The MS$^2$ AGC target value was set to $10^5$ with a maximum IT of 50 ms.

### Protein identification

Raw files were analysed using Proteome Discoverer with the Sequest (Thermo Fisher Scientific, San Jose, USA; version 2.5). Sequest was set up to search against UniProt human database (retrieved in November 2019), common contaminants and sequence of MCAK protein with trypsin as the digestion enzyme. A fragment ion mass tolerance was set to 0.02 Da and a parent ion mass tolerance to 10 ppm. The number of maximal allowed missed cleavages was set to 3. Carbamidomethylation of cysteine was specified as a fixed modification, deamidation of asparagine and glutamine, oxidation of methionine, GG-modification of lysine and acetylation, loss of methionine and loss of methionine plus acetylation of the protein N-terminus were specified as variable modifications. Abundances were calculated as intensities.

## IP and Western blot analysis

For co-IP, cells from 1 to 4 confluent 15-cm dishes were harvested using either a cell lifter or trypsinisation and lysed in 500 μl ice-cold lysis buffer (20 mM HEPES pH 7.3, 110 mM K-acetate, 2 mM Mg-acetate, 1 mM EGTA, 0.2% NP-40 (for Flag-IP) or 0.4% (for Fbxw5-IP), 1 μg/ml aprotinin/pepstatin/leupeptin, 1 mM Pefabloc, 2 mM Na-Orthovanadate, 5 mM NaF). Lysates were cleared by centrifugation at 20,000 $g$ for 30 min. For Flag-Fbxw5 IPs, 10 μl FLAG-agarose was added for 3 h under constant agitation. For Fbxw5 IP, 1.5 μg anti-Fbxw5 IgG or unspecific rabbit IgG was added for 1 h under constant agitation, followed by addition of 10 μl pre-equilibrated Protein A agarose for 2 h. Beads were washed four times with lysis buffer, and for Flag-IP in Fig EV1C, proteins were eluted by three consecutive elutions at 25°C under constant agitation with three bead volumes of elution buffer (lysis buffer supplemented with 150 mM NaCl and 0.2 mg/ml FLAG-peptide). Eluates were combined and precipitated with 10% trichloroacetic acid. Precipitates were washed once with −20°C cold acetone, air-dried and resuspended in 50 μl 1× SDS sample buffer (50 mM Tris pH 6.8, 1% (w/v) SDS, 0.1% (w/v) bromophenol blue, 10% glycerol, 100 mM DTT). For all other IPs, beads were washed four times with lysis buffer and proteins were eluted in 40 μl 1× sample buffer.

Since Fbxw5 signals were undistinguishable from background signals upon direct lysis in sample buffer, cells were lysed in 35 μl ice-cold lysis buffer followed by clearance at 20,000 $g$ for 10 min at 4°C. Afterwards, protein concentrations were measured using Bradford assay, adjusted and 30 μl were mixed with 10 μl 4× sample buffer. This procedure was used for all straight Western blots shown.

For Western blotting, proteins were resolved on 7.5% or 7.5–15% gradient SDS–polyacrylamide gels. After electro-transfer onto PVDF membranes using wet-tank blotting systems (self-made), membranes were stained with Coomassie (0.1% Coomassie Brilliant Blue R250 in 50% ethanol and 10% acidic acid) and cut in order to simultaneously detect two proteins from the same blot. After destaining (50% ethanol, 10% acidic acid) and washing with $H_2O$, membranes were blocked in PBS-T (140 mM NaCl, 2.7 mM KCl, 10 mM $Na_2HPO_4$, 1.5 mM $KH_2PO_4$, pH 7.5, 0.1% Tween-20) containing 5% skimmed milk powder (Roth) for 2 h or overnight. Primary antibodies were diluted in PBS-T with 5% milk powder and incubated for 2 h or overnight. After washing with PBS-T, horseradish peroxidase-conjugated secondary antibodies (Jackson ImmunoResearch/Invitrogen) were added and proteins were detected with Immobilon Western chemiluminescent HRP substrate and a Fujifilm LAS-4000 Luminescence Image Analyser. In order to detect other proteins on the same blot, blots were incubated overnight with the primary antibody of the other protein (different species) together with ~ 10 mM $NaN_3$ in order to quench signals from the first secondary antibody.

## Pull-down experiments

For *in vitro* pull-down experiments, purified proteins were mixed in a molar ratio of 5 : 1 (prey vs bait) together with competing *E. coli* lysate in PD Buffer (20 mM HEPES pH 7.3, 110 mM K-acetate, 2 mM Mg-acetate, 1 mM EGTA, 300 mM NaCl, 20 mM imidazole, 0.1% NP-40, 1 μg/ml aprotinin/pepstatin/leupeptin) and incubated with Ni-NTA agarose for 1 h. After 4× washing with

PD Buffer, proteins were eluted in 2× sample buffer containing 200 mM imidazole.

## Immunofluorescence

24 h after siRNA transfection, cells were seeded onto 12-mm-diameter glass coverslips (Neolab). Cells were fixed in ice-cold methanol for at least 10 min, rehydrated in PBS for 10 min and blocked for 60 min in IF blocking buffer (PBS-T with 2% FBS, 1% BSA, filtered 0.45 μm). Cells were incubated with primary antibodies diluted in IF blocking buffer in a dark and humid chamber for 2 h. After 3× washing with PBS + 0.1% Tween-20 for 5 min, cells were incubated with Alexa-conjugated secondary antibodies (either anti-mouse or anti-rabbit conjugated either to Alexa 488 or Alexa 594, respectively) diluted in IF blocking buffer for 1.5 h in a dark and humid chamber. Hoechst (20 μg/ml) was included with the secondary antibody to stain DNA. After three washes with PBS-T and one wash with PBS, slides were mounted on glass slides (Thermo Scientific) using fluorescence mounting medium. Images (five stacks 0.5 μm apart) were taken using either a Nikon Ni-E upright microscope equipped with an air objective (Nikon Plan Apo λ 40× NA 0.95) and a DS-Qi2 black and white camera (Figs 7A, EV3B and C, EV4C and F, and EV5B and C) or with a Zeiss Axio Observer fluorescence microscope equipped with oil immersion objectives (LD-Plan-Neofluar 40×/1.3 or Plan-Apochromat 63×/1.4 [Zeiss]), Colibri LED light source, AxioCam MRm camera (all other images).

## Live-cell imaging

RPE-1 cells expressing mNG-MCAK under a doxycycline-inducible promoter were transfected with siRNAs, seeded onto Ibidi 8 Well Glass Bottom μ-slides and induced by adding doxycycline. For pulse-chase experiment, 24 h post-induction and 48 h post-transfection, cells were washed 2× with PBS and incubated in medium without FBS and doxycycline. Cells were subsequently imaged with a Leica DMi8 spinning disk microscope at 40× or 63× magnification (HC PL APO 40×/1.3 Oil or HC PL APO 63×/1.40–0.60 Oil) equipped with a Hamamatsu Orca Flash 4.0 LT. Imaging was performed at 37°C and 5% humidified $CO_2$. Telophase cells were chosen for time-lapse imaging and 13 stacks at a distance of 0.5 μm were taken every 20 min. If not enough telophase cells were present, cells in prometaphase and metaphase were imaged and quantification started at telophase.

## FRAP analysis

For fluorescence recovery after photobleaching (FRAP), a Zeiss LSM 780 microscope with a 63×/1.40 NA Plan-Apochromat oil objective lens (Zeiss) and a 488 nm Argon laser was used. A small ~ 2 μm$^2$ region of interest (ROI) was photobleached and fluorescence recovery was monitored over time, as described previously (Lippincott-Schwartz & Patterson, 2003). FRAP images shown were processed with a 1-pixel-radius linear mean filter for better visualisation.

## Image analysis

Image analysis was performed using FIJI (Schindelin et al, 2012). All immunofluorescence images show maximum intensity projections of 5 z-stacks (each 0.5 μm apart) with equal min/max display settings for all images of one experiment. For analysis of MCAK signals at ODF in Fig 4A, total ROIs were generated using a threshold at 200 followed by particle analysis. Total signal intensity of ROIs overlapping with ODF2 signals was measured within maximum intensity projections of the MCAK channel and subtracted by a mean background intensity of a cell-free area multiplied with the area of the corresponding ROI. Analysis of MCAK signals at ODF2 in Figs 5D and EV4D was performed accordingly, except that here ROIs were generated by thresholding after subtracting two maximum intensity projection images subjected to two different Gaussian blurs ($\sigma = 2$ or $3$ + threshold 25 (Fig 5D) and $\sigma = 3$ or $4$ + threshold 3 (Fig EV3B)) in order to enable use of same settings for images with highly diverse MCAK signal intensities.

For analysis of time-lapse images, centrosomal mNG-MCAK signals were measured on maximum intensity projections for each time point within a circular ROI (diameter 11 pixels). For background subtraction, extracellular background was measured with the same ROI and the average background intensity for three representative movies per experiment was subtracted from the mNG-MCAK centrosomal signal intensity value.

## NanoBRET™ assay

Hela cells were transfected with 2 μg of HaloTag® fusion construct and 20 ng NanoLuc® fusion construct using Fugene® HD (Promega) according to the manufacturer's instructions. Approximately 20 h later, the cells were harvested, resuspended in NanoBRET™ assay medium (phenol red-free Opti-MEM™ with 4% FCS) and seeded at $2 \times 10^4$ cells/well into white, 96-well flat-bottom tissue culture-treated plates (Greiner) in the absence (DMSO vehicle control) or presence of 100 nM HaloTag® NanoBRET™ 618 Ligand. Immediately before measurement, the NanoBRET™ Nano-Glo® substrate (Promega) was added at 10 μM. Luminescence signal was detected at a wavelength range of 415–485 nm for the donor and 610–700 nm for acceptor signals with the Tecan Spark 10 M plate reader. NanoBRET™ ratio (BU) was calculated by dividing the acceptor signal by the donor signal. The corrected NanoBRET™ ratio was calculated by subtracting the DMSO vehicle control sample from the experimental sample with cells treated with HaloTag® NanoBRET™ 618 Ligand.

## Incucyte

Cells were seeded on 12-well plates 24 h before imaging with an Incucyte® S3 Live-Cell Analysis System. Images were taken using the 20× air objective. H2B-mRuby signals were used to count all cells automatically using the same settings for all images via the Incucyte software. Mitotic cells were counted manually, and the percentage was afterwards calculated.

## Plasmid generation

All plasmids used in this study are shown in Appendix Table S1 and were generated by standard cloning techniques either via site-directed mutagenesis, Gibson assembly (Gibson et al, 2009) or restriction enzyme-based strategies (New England Biolabs) using E. coli strain DH5α. Oligonucleotides for PCR amplification were

obtained from Sigma-Aldrich/Merck. For each construct, the relevant open reading frame was completely verified by Sanger sequencing (GATC/Eurofins Scientific). Plasmid name indicates vector backbone followed by gene of interest with or without tag or mutations. All genes are of human origin, except for plasmids pBigBac-HisCul4A/His-DDB1/His-mmRbx1 and pBigBac-HisCul1/His-mmRbx1, in which Rbx1 is derived from *Mus musculus*. pFastBac1-Skp1ΔΔ contains human Skp1 with deletion of two loops (aa 38–43 and 71–82) (Schulman *et al*, 2000). pBigBac-HisCul4A/His-DDB1/His-mmRbx1 contains Cul4A (aa 38–759), DDB1 (aa 1–1,140) and mmRbx1 (aa 12–108) (Fischer *et al*, 2011). pBigBac-HisCul1/His-mmRbx1 contains full-length Cul1 and mmRbx1 (12–108). pcDNA3.1-HA-CSPP1 contains CSPP1 (878–1,256). All constructs generated in this work contain the corresponding canonical sequence (according to http://www.uniprot.org), except for Kif2a constructs (isoform 1, identifier: O00139-1) and ODF2 constructs (isoform 3, identifier: Q5BJF6-3).

## Data availability

The mass spectrometry proteomics data have been deposited to the ProteomeXchange Consortium via the PRIDE (Perez-Riverol *et al*, 2019) partner repository with the dataset identifier PXD026353 (http://www.ebi.ac.uk/pride/archive/projects/PXD026353). The data of the protein microarray are presented in Dataset EV1.

**Expanded View** for this article is available online.

## Acknowledgements

This project received funding from the Deutsche Forschungsgemeinschaft (SPP1365, ME 2279/3 to F.M. and DFG KN590/7-1 to K-P.K), the European Union's Horizon 2020 research and innovation programme under the Marie Skłodowska-Curie Grant Agreement No. 748315 (UBIMAPS) and the Heidelberg CellNetworks Cluster of Excellence postdoctoral programme (to J.S.). We would like to thank Vladimir Benes for providing access to a microarray reader and Brenda Schulman for the generous gift of recombinant neddylated Cul1/Rbx1 and plasmids of neddylation enzymes. We gratefully acknowledge Giorgio Scita, Eric Fischer, Ludger Hengst, Ingrid Hoffmann, Achim Dickmanns, Sima Lev, Shahri Raasi, Elmar Schiebel, Gislene Pereira and Bahtiyar Kurtulmus for plasmids and advice, and thank Holger Lorenz, Christian Hörth, Monika Langlotz, Thomas Ruppert and Sabine Merker for excellent microscopy, FACS and mass spectrometry support, respectively. Finally, we would like to thank Anthony Razov, Hannah Lee, Lisa Mutz and Moritz Dodenhöft for technical support, Annette Flotho for critical reading of the manuscript and the whole Melchior laboratory for helpful discussions, reagents and advice. Open Access funding enabled and organized by Projekt DEAL.

## Author contributions

JS designed and performed most experiments and wrote the manuscript. GH designed and performed the protoarray screen and wrote this part of the manuscript. SHe designed and performed the BRET assays. FMi designed and performed the post-mitotic degradation assays (Figs 5A and EV4D). RB conducted the mass spectrometry experiment. SHa generated the RPE-1 TET3G cell line. KM purified various proteins and carried out some ubiquitylation assays. K-PK designed the BRET assays, and FMe guided the project, designed experiments and wrote the manuscript.

## Conflict of interest

The authors declare that they have no conflict of interest.

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
