## [Review Process File · The EMBO Journal]

SCFFbxw5 targets kinesin-13 proteins to facilitate ciliogenesis

Jörg Schweiggert, Gregor Habeck, Sandra Hess, Felix Mikus, Roman Beloshistov, Klaus Meese, Shoji Hata, Klaus-Peter Knobloch, and Frauke Melchior

DOI: [10.15252/embj.2021107735](https://doi.org/10.15252/embj.2021107735)

Corresponding author(s): [Frauke Melchior \(f.melchior@zmbh.uni-heidelberg.de\)](mailto:f.melchior@zmbh.uni-heidelberg.de), [Jörg Schweiggert \(j.schweiggert@zmbh.uni-heidelberg.de\)](mailto:j.schweiggert@zmbh.uni-heidelberg.de)

Review Timeline:

Submission Date:	14th Jan 21
Editorial Decision:	23rd Feb 21
Revision Received:	30th May 21
Editorial Decision:	22nd Jun 21
Revision Received:	1st July 21
Accepted:	12th Jul 21

Editor: *Ieva Gailite*

Transaction Report:

Thank you for submitting your manuscript for consideration by The EMBO Journal. We have now received three referee reports on your manuscript, which are included below for your information.

As you will see from the comments, all reviewers appreciate the study and find the presented mechanism of Fbxw5-dependent degradation of MCAK and its role in ciliogenesis interesting, they also indicate a number of concerns that would have to be addressed and clarified before they can support publication of the manuscript, in particular asking to strengthen the insights into the molecular details of Fbxw5/MCAK interaction, including at the endogenous level, and to clarify the observed temporal delay between MCAK regulation and ciliogenesis defects during the cell cycle. Therefore, I would like to invite you to address the concerns raised by all reviewers in a revised

REFEREE REPORTS

Referee #1:

In the manuscript the authors identify a novel substrate for the SCFFbxw5 E3 ubiquitin ligase. The authors use a novel techniques for substrate identification based on protein microarrays. The technique reveals known and novel substrates and among them the microtubule regulating proteins MCAK. The authors establish that MCAK specifically interacts with Fbxw5 and show that Fbxw5 is able to ubiquitylate MCAK in vitro through K48 ubiquitylation. Furthermore, it is shown that Fbxw5 can interact with MCAK in vivo through a nanobret assay. The authors present evidence that MCAK is also targeted for ubiquitylation and degradation by APC/C Cdh1 during G1 while Fbxw5 targets MCAK in G2/M. The biological significance of MCAK ubiquitylation is reconducted to a control mechanism regulating ciliogenesis in G0. If Fbxw5 is depleted, MCAK accumulates leading to defective ciliogenesis in next G0.

The biochemical experiments are elegant and well conducted. I was pleased to see a description of the results that was rational and very well detailed. Overall, the manuscript was well written, structured.

1. It would have been nice to see some analysis of the degron recognised by Fbxw5. Is there anything in common among all the proteins identified on the microarray which allows Fbxw5 interaction?
2. I have one major concern and I would appreciate if the authors would clarify this point and add some explanations in the discussion section. The temporal degradation of MCAK is such that the proteins not degraded in G2 would be taken over by APC/C. Why is there a phenotype in G0 associated to the loss of Fbxw5?

Other minor concerns are below:

1. I suggest to add a Wb of Fbxw5 when cells are treated with Doxycycline to compare expression to endogenous protein.

Referee #2:

In this manuscript, Schweiggert and colleagues employ a protein microarray screen to identify substrates of the SCF-FBXW5 ubiquitin ligase. Among 161 candidate substrates, they show that FBXW5 interacts with, and mediates the ubiquitylation of, the microtubule depolymerase Kif2C/MCAK as well as its orthologs Kif2A and Kif2B. The authors demonstrate that the FBXW5-dependent ubiquitylation of MCAK leads to its proteasomal degradation during G2/M and is required for the formation of primary cilia in the following G1.

In my opinion this study is potentially interesting, however, additional data is needed to fully support the authors' model. Below I provide ways this study can be strengthened.

The FBXW5-MCAK interaction in cultured cells is assessed by coimmunoprecipitation of overexpressed proteins (Figure 2). Does overexpressed FLAG-tagged FBXW5 coimmunoprecipitate with endogenous MCAK? Does overexpressed HA-tagged MCAK coimmunoprecipitate with endogenous FBXW5? Is a complex with the endogenous FBXW5 and MCAK proteins detected?

FBXWs interact with their substrates via their WD40 repeats. Have the authors tested whether FBXW5 WD40 mutants are able to bind and ubiquitylate endogenous MCAK? Conversely, have the authors identified the FBXW5-binding region in MCAK?

Is MCAK ubiquitylated in cultured cells? If so, is MCAK ubiquitylation in cells dependent on FBXW5 expression?

The data about the mechanism by which FBXW5-dependent degradation of MCAK controls ciliogenesis is interesting but limited. Have the authors investigated the effect of expressing physiological levels of a non-degradable MCAK mutant (unable to bind FBXW5) in cells?

Minor points:

Figure 4A. To have a clearer picture of MCAK levels during cell cycle, the authors should synchronize, release from the synchronization block and collect cells at different time points. Such a time course would provide more information about the oscillation of MCAK during cell cycle that is only hinted in this figure.

The authors state: "MCAK amounts were slightly increased in asynchronously growing cells", however no difference is detected in the levels of MCAK in figure 4A. I do not see correspondence between the levels of MCAK in asynchronous cells in the blot in figure 4A and the ones in graph 4B, even considering the ratio with the levels of the loading control UBA2. I understand that the graph represents the average of 4 independent experiments, but can the authors show a more representative blot?

Figure 4A. The band corresponding to FBXW5 is barely visible. The authors should show a longer exposure FBXW5 blot in addition to the one already present.

Figure 3b. Do the "only Ubch5b" and "only Cdc34" samples contain also components of the SCF-FBXW5 complex? If so, specify it in the legend.

Figure 1A. The calculated size of Cul1 size is about 89 kDa and it usually runs on SDS-PAGE at 70-80 kDa, however, in this figure it runs at about 50 kDa. Are they truncated forms? If so, explain.

Figure 2C. Additional negative controls, such as for instance HT-FBXW7 (or any other FBXW) with MCAK-NL, are needed.

Referee #3:

Review of SCFFbxw5 targets MCAK in G2/M to facilitate ciliogenesis in the following cell cycle by Schweiggert et al.

This manuscript identifies the microtubule destabilising kinesin, MCAK, as a substrate of the SCFFbxw5 E3 ligase. Briefly, in search of new SCFFbxw5 substrates the authors performed an in vitro ubiquitylation screen on a protein array, and isolated 161 candidate proteins that included known and new substrates such as MCAK. The report convincingly demonstrates that MCAK can be ubiquitylated by this E3 ligase and also that this modification predominantly occurs via insertion of K48-linked ubiquitin chains at multiple sites within MCAK.

Having isolated MCAK as a substrate of SCFFbxw5, the authors went on to investigate the purpose of this modification, and carried out a number of cell synchronisation, siRNA and protein overexpression studies to this end. Their key conclusion is that SCFFbxw5-dependent ubiquitylation of MCAK during G2 is important for ciliogenesis in G0. In particular, the authors find that depletion of SCFFbxw5 or exogenous expression of MCAK in serum-starved cells preclude cilia assembly. Co-depletion of MCAK and SCFFbxw5 restores normal ciliogenesis, indicating that in absence of MCAK, SCFFbxw5 is dispensable for cilia formation; this piece of data represents the most compelling evidence that SCFFbxw5-dependent degradation of MCAK promotes cilia assembly.

Overall, this manuscript is both interesting and informative. However, the conclusion that degradation of MCAK by SCFFbxw5 occurs in G2 with the outcome manifesting in ciliogenesis during the next cell cycle needs further experimental support.

Specific points:

1. The impact of Fbxw5 depletion on MCAK levels is shown on a western blot in Fig 4A. Corresponding quantitation in Fig 4B compares signal intensities by normalising to the control of

each condition. I do find it unusual to show the fold differences in the same graph when the controls are different. It may also be more informative to show the data normalised against Uba2 only, as this would enable readers to compare effects of different treatments. For example, the impact of nocodazole on MCAK levels is much higher than any other treatment, yet this is not apparent from the graph. The authors may also want to consider changing presentation of the graph in Fig 6B.

2. Fig.4C shows very clearly that centrosomal MCAK levels in serum-starved cells are affected by Fbxw5 depletion. It is a compelling idea that SCFFbxw5 may target a specific centrosomal pool of MCAK. A caveat here is that due to its multiple substrates SCFFbxw5 depletion is likely to have pleiotropic effects on the cell cycle and even in centrosomes, making it difficult to conclude that centrosomal increase in MCAK is solely due to depletion of this E3 ligase. The manuscript could be made much more impactful if lysine-mutant MCAK was generated that is resistant to SCFFbxw5. SCFFbxw5 -resistant MCAK expressed as a transgene or a gene edited version would enable more specific functional studies.

3. It is not easy to reconcile findings that degradation of MCAK by SCFFbxw5 in G2 is required for ciliogenesis in the next G0. What makes this a difficult model is that MCAK levels peak in mitosis; this would suggest that the MCAK pool that is degraded in G2 would not be replenished by mitotic MCAK and that this pool is also inaccessible to APC/C. Certain centrosomal structures such as subdistal and distal appendages are removed/remodelled when cells enter mitosis, and could be candidates for such pools. The authors could investigate where exactly MCAK (endogenous and overexpressed, or upon Fbxw5 depletion) localises within the centrosome in G2, M and in G0 using super-resolution or expansion microscopy.

4. Could the authors test if Fbxw5 localises to centrosome in G2 but not in G0? This would support the model propose. Also, can the authors exclude that the massive reduction in MCAK levels in Fbxw5-depleted and RO3306-treated cells is due to inhibition of CDK1 rather than the G2 arrest?

5. The authors should include western blots of mNG-MCAK overexpressing cells together with controls blotted with MCAK antibodies, so that readers can appreciate extent of overexpression. Immunofluorescence with MCAK antibodies would also be helpful.

Step by step response to reviewers:

Referee #1:

In the manuscript the authors identify a novel substrate for the SCFFbxw5 E3 ubiquitin ligase. The authors use a novel techniques for substrate identification based on protein microarrays. The technique reveals known and novel substrates and among them the microtubule regulating proteins MCAK. The authors establish that MCAK specifically interacts with Fbxw5 and show that Fbxw5 is able to ubiquitylate MCAK in vitro through K48 ubiquitylation. Furthermore, it is shown that Fbxw5 can interact with MCAK in vivo through a nanobret assay. The authors present evidence that MCAK is also targeted for ubiquitylation and degradation by APC/C Cdh1 during G1 while Fbxw5 targets MCAK in G2/M. The biological significance of MCAK

ubiquitylation is reconducted to a control mechanism regulating ciliogenesis in G0. If Fbxw5 is depleted, MCAK accumulates leading to defective ciliogenesis in next G0.

The biochemical experiments are elegant and well conducted. I was pleased to see a description of the results that was rational and very well detailed. Overall, the manuscript was well written, structured.

We thank the reviewer for his/her very positive reply and hope that our comments and the additional data dissolve his/her remaining concerns.

1. It would have been nice to see some analysis of the degron recognised by Fbxw5. Is there anything in common among all the proteins identified on the microarray which allows Fbxw5 interaction?

This is an interesting question and we tried to identify a general Fbxw5 degron via different motif search algorithms (Dilimot, STREME, MEME, GLAM2) as well as by engaging expert bioinformaticians. Unfortunately, we were unable to find a motif that is enriched in the Fbxw5 substrate set. So far only one degron for an Fbxw5 substrate has been identified: Jeong et al. identified a peptide spanning residues 180-194 of Sec23b as an interaction site for Fbxw5. Phosphorylation of S186 within this peptide by ULK1 negatively regulates the interaction and a phosphomimetic Sec23b S186D mutant was stabilised (Jeong et al., eLife, 2018). The notion of substrate recruitment that does not require a preceding phosphorylation is supported by our own data on Eps8 (Werner et al., 2013) as well as Kif2c/MCAK, Kif2a and Kif2b (Figure 3F, EV2J). ULK1 consensus sites (Egan, Mol Cell 2015) are present in the primary sequence of MCAK, Eps8 and 56 other protoarray candidates but not in Sas6. However, ULK1 consensus sites are equally abundant in a control set of proteins not ubiquitylated on the array. The ULK1 consensus site consists of a serin surrounded by aliphatic and aromatic residues (Egan, Mol Cell, 2015). Notably, our own data also point to an interaction of Fbxw5 and Eps8 via aromatic and aliphatic residues, but so far we were not able to identify a corresponding motif within MCAK (see also below, answer to reviewer 2). We initiated a collaboration with a structural biology lab to solve structures of Fbxw5 in complex with MCAK to further work on this interesting aspect, but we hope the reviewer agrees that this is beyond the scope of this manuscript. Nevertheless, we believe that this is an important question, which will be of interest for a broader audience and we therefore included a corresponding paragraph into the discussion section of our manuscript.

2. I have one major concern and I would appreciate if the authors would clarify this point and add some explanations in the discussion section. The temporal degradation of MCAK is such that the proteins not degraded in G2 would be taken over by APC/C. Why is there a phenotype in G0 associated to the loss of Fbxw5?

This is an important aspect and we thank the reviewer for his comment, as it shows us that we have not sufficiently explained this issue. The kinetics of the post-mitotic degradation of MCAK via the APC/C can be appreciated in Figure 5A, in which MCAK is removed from centrosomes with a half-life of about 6 hours. Fbxw5-

depleted cells show approximately 2-fold higher MCAK amounts at time point zero and thus need an additional ~6 hours within G₀ to reach the levels of unperturbed cells at mitotic exit. This is a massive delay of 1/4 of the total time spent in serum-free medium (i.e. 24 hours) in our ciliogenesis assays. Taking into account that the percentage of ciliated RPE-1 cells reaches a plateau already after 8 hours in serum starvation (Kurtulmus et al., 2018), we thus believe that the APC/C is simply not fast enough to reach sufficiently low MCAK levels in time. In order to better clarify this issue, we added the following part into our discussion:

“Our time-lapse analysis indicates that the APC/C-dependent degradation of MCAK after mitosis is rather slow, with a half-life of about 6 hours (Fig 5A). Thus, cells deficient of Fbxw5 would reach MCAK levels of unperturbed cells only after 6 hours, which depicts a massive delay considering that RPE-1 cells form cilia already after 8 hours in serum starvation⁶⁷. Taking into account the concomitant increase in the amounts of Kif2a and Kif2b, this can explain why the APC/C is apparently unable to compensate for loss of Fbxw5.”

Other minor concerns are below:

1. I suggest to add a Wb of Fbxw5 when cells are treated with Doxycycline to compare expression to endogenous protein.

We thank the reviewer for this important comment. As outlined below, too much overexpression was indeed an issue in one of the experiments.

We now included WB and IF images of cells used for Figure 5A (same doxycycline induction) in Figure EV4B and EV4C. They show a modest increase compared to endogenous MCAK. Please note that due to a shortage of our monoclonal anti-MCAK antibody from Santa Cruz, we used a polyclonal rabbit anti-MCAK antibody from Novus Biologicals (NB100-2588) for some of the new figures (indicated in the key resource table). We validated the specificity of this antibody in new Figure EV4A.

We also included Figure EV4F showing WB (various doxycycline concentrations) and IF images (10 ng/ml doxycycline as in Figure EV4E) under full serum conditions of cells used for Figure EV3E, EV4E (= old Figure EV3C), Figure 7A and B, EV5B and C, as well as old Figure 7A and old Figure EV4B, C and D. Here, the extent of MCAK overexpression was a bit stronger but still reasonable at least for cells in full serum.

What we unfortunately had failed to consider was the decline of MCAK upon serum starvation. As we had kept doxycycline present throughout serum starvation, exogenous expression continued to increase. Together, this led to a drastic fold-difference between ectopic and endogenous MCAK 24 and 48 hours after serum starvation (see Figure 1 for reviewers attention).

Figure 1 for reviewers attention. Immunoblot of extracts of RPE-1 cells used for old Figure 7A and old Figure EV4B and C.

Although we still believe that this experiment provides potentially interesting insights, we decided to remove these panels and the corresponding parts in the results and discussion section from our manuscript. Instead, we included now an experiment where we first induced mNG-MCAK overexpression in full serum for 24 hours, followed by washout of doxycycline at the time of serum withdrawal (new Figure 7A, new Figure EV5B and C). Here, the extent of overexpression was not as drastic (new Figure 7B). Although the effect on ciliogenesis was also much lower, we still observed a significant reduction in ciliated cells and shortening of remaining cilia that correlated with mNG-MCAK levels at basal bodies.

The difference in cilia reduction between moderate MCAK overexpression (mild effect) and Fbxw5 knockdown (strong effect) prompted us to ask whether additional Fbxw5 targets may contribute to the strong effect. Obvious candidates were MCAK's orthologs Kif2a and Kif2b. In line with our *in vitro* data shown in the initial manuscript, we now found that levels of endogenous Kif2a and ectopically expressed mNG-Kif2b were also increased upon Fbxw5-depletion under serum-starvation (Figure EV5E and F). Interestingly, knockdown of Kif2a and Kif2b could also rescue the Fbxw5-dependent ciliogenesis defect, although not as efficiently as MCAK (Figure EV5G and H). Together, these data suggest that the ciliogenesis defect upon Fbxw5 depletion is due to a combined increase in kinesin-13 activity, which explains the difference between ciliogenesis reduction between MCAK overexpression and Fbxw5 knockdown. We decided to adjust the title and abstract of our manuscript to accommodate these findings.

We would like to thank the reviewer for bringing up this important point, as it significantly improved both the quality and scope of our manuscript.

Referee #2:

In this manuscript, Schweiggert and colleagues employ a protein microarray screen to identify substrates of the SCF-FBXW5 ubiquitin ligase. Among 161 candidate substrates, they show that FBXW5 interacts with, and mediates the ubiquitylation of, the microtubule depolymerase Kif2C/MCAK as well as its orthologs Kif2A and Kif2B. The authors demonstrate that the FBXW5-dependent ubiquitylation of MCAK leads to its proteasomal degradation during G2/M and is required for the formation of primary cilia in the following G1.

In my opinion this study is potentially interesting, however, additional data is needed to fully support the authors' model. Below I provide ways this study can be strengthened.

We thank the reviewer for his/her suggestions and hope that our comments and the additional data satisfy his/her concerns.

The FBXW5-MCAK interaction in cultured cells is assessed by coimmunoprecipitation of overexpressed proteins (Figure 2). Does overexpressed FLAG-tagged FBXW5 coimmunoprecipitate with endogenous MCAK? Does overexpressed HA-tagged MCAK coimmunoprecipitate with endogenous FBXW5? Is a complex with the endogenous FBXW5 and MCAK proteins detected?

We performed co-immunoprecipitation experiments using Fbxw5-directed antibodies under different cell cycle arrest conditions, and reproducibly observed an interaction between endogenous Fbxw5 and MCAK upon nocodazole arrest (Figure 2D). This could indicate that the interaction is particularly enhanced during mitosis, but it is of course also possible that the increased amounts of MCAK (as seen in the input samples) are partially responsible for this result. Please note, Fbxw5 is very low in abundance, which makes co-IPs particularly challenging (e.g., the proteomics study Beck et al 2014 indicates less than 500 copies per cell). Considering our combined data on the interaction between endogenous proteins, purified proteins and within intact cells, we believe to provide sufficient data for our claim that Fbxw5 is able to recruit MCAK.

FBXWs interact with their substrates via their WD40 repeats. Have the authors tested whether FBXW5 WD40 mutants are able to bind and ubiquitylate endogenous MCAK? Conversely, have the authors identified the FBXW5-binding region in MCAK?

Since we are not aware of specific and functional point mutations within the WD40 repeats of Fbxw5, the only way to test their involvement in the binding of MCAK would be via truncations. However, structural prediction of Fbxw5 by the Phyre2 engine indicates that the whole region C-terminal to the F-box domain folds into a seven-bladed beta-propeller (see Figure 2A for reviewers attention).

Figure 2 for reviewers attention. Phyre2 prediction of Fbxw5 structure. The N-terminal alpha-helical domain represents the F-box domain, the beta-propeller the remaining C-terminal part **A.** Top view. The two C-terminal WD40 motifs are highlighted in red. **B.** Side view. Exposed aromatic residues are highlighted in red.

Even if we would delete only the two C-terminal WD40 domains (and leave the others intact), this would probably destroy the whole fold making it difficult to draw conclusions. We therefore tried to identify specific amino acid residues involved in substrate recruitment based on data for two other Fbxw5 targets, Eps8 and Sec23b. Earlier work from our lab on Eps8 based on truncations and point mutation analysis has shown that a stretch of aromatic and aliphatic residues within Eps8 could be involved in binding Fbxw5 (unpublished data, not shown). This is in line with the motif identified by Jeong et al., in which the ULK1-targeted Ser is followed by bulky hydrophobic residues Tyr, Val and Phe. Based on this, we searched for surface exposed aromatic residues on the predicted Fbxw5 structure that may be involved in binding bulky hydrophobic residues and identified F189 and Y192 as interesting candidates. Mutation of these residues to alanine did indeed abrogate the interaction between Fbxw5 and Eps8 as well as between Fbxw5 and Sec23b. However, it did neither affect binding towards HGS nor MCAK (see Figure 3A and B for reviewers attention), suggesting that the mechanism for substrate recruitment of Fbxw5 is more complex and varies between different targets.

Figure 3 for reviewers attention. A. Co-IP between Flag-Fbxw5 and Eps8, Sec23b and HGS. **B.** Co-IP between Flag-Fbxw5 and HA-MCAK

Regarding the identification of binding regions on MCAK, we carried out co-IP experiments of MCAK truncations. These assays suggested that the motor domain of MCAK (aa 231-583) alone is sufficient to promote interaction with Fbxw5 (see Figure 4A for reviewers attention), which makes sense considering that this region is highly conserved among our identified substrates Kif2a, Kif2b and MCAK.

Figure 4 for reviewers attention. A. Anti-HA IP of different HA-MCAK truncations with Flag-Fbxw5. **B.** Anti-Flag IP of Flag-Fbxw5 with different HA-MCAK motor domain truncations. **C.** Anti-Flag IP of Flag-Fbxw5 with HA-MCAK full length and HA-MCAK carrying an internal deletion of amino acids 359-499

However, we were not able to further narrow down the interaction site, as further truncations displayed strongly different expression patterns and although amino acids 359 to 499 efficiently precipitated with Fbxw5, an internal deletion of these residues within full length MCAK did not affect the binding to Fbxw5. Besides the possibility of multiple binding sites, other explanations for this observation could be that taken out of the context of the full length protein, residues become exposed that make the whole polypeptide in general sticky or that, in the context of the full length protein, dimerization with endogenous MCAK still facilitates the binding even if the corresponding region is deleted. Since we were not able to generate these MCAK truncations recombinantly in bacteria, at least in a quality comparable to full length MCAK, we decided to omit these data from our manuscript as we consider them as non-conclusive. As already mentioned in our response to reviewer #1, we recently initiated a collaboration with a structural biology lab to hopefully solve the structure of Fbxw5 in complex with MCAK, but we hope the reviewer agrees that this is beyond the scope of this manuscript.

Is MCAK ubiquitylated in cultured cells? If so, is MCAK ubiquitylation in cells dependent on FBXW5 expression?

Yes indeed, there is actually strong evidence in the literature that MCAK is ubiquitylated endogenously in cells, which we now clarify better in the revised version. Proteomic studies have identified 27 ubiquitylation sites of MCAK (PhosphoSitePlus – www.phosphosite.org, Akimov et al., 2018, Udeshi et al., 2013). To better compare our approach with the available data, we carried out mass spectrometry analysis of diGly containing peptides within our *in vitro* ubiquitylation experiment (see also reviewer #3 question 2) and identified 18 lysine residues that are modified by SCF^{Fbxw5} in concert with Cdc34 (Fig EV2M). Importantly, 15 out of these 18 lysines have been also annotated as ubiquitylation sites in cultured cells within the above mentioned studies. Since MCAK contains 54 lysine residues in total of which 27 are annotated at PhosphoSitePlus, this represents a significant enrichment and underlines the physiological relevance of our *in vitro* approach.

Visualisation of ubiquitylated species is commonly done by purification of His-tagged ubiquitin under denaturing conditions followed by immunoblotting. However, this approach is not applicable in our case, since our *in vitro* ubiquitylation experiments clearly demonstrate that a His-tag on ubiquitin drastically impairs MCAK ubiquitylation by SCF^{Fbxw5} (Figure EV2C, D and E). Furthermore, comparing the amounts of ubiquitylated species within cells as readout for E3 ligase activity has severe pitfalls, especially if the levels of the target protein are affected. In our case, the higher amounts of MCAK upon Fbxw5 depletion could lead to an indirect increase in its ubiquitylated species by other E3 ligases (e.g. the APC/C) making the interpretation of such experiments difficult.

We thus believe that our comprehensive *in vitro* approach (Figure 3 and Figure EV2) is much more meaningful, because it unambiguously shows that MCAK is both efficiently and specifically targeted by SCF^{Fbxw5} in a direct manner. Our new data now demonstrate that the sites ubiquitylated here have been found to be also modified within cells, which further emphasizes the biological relevance of our approach. This is well complemented by our comprehensive binding studies (Figure 2), the various stabilisation assays (Figure 4C, Figure 7C and Figure EV4D) and finally the new

CHX chase experiment in synchronised cells (Figure 6B, see also answer to reviewer #3, specific point 4) that do not suffer from the above mentioned issues as each sample is compared to the first time point, making the results much more independent of varying input amounts.

The data about the mechanism by which FBXW5-dependent degradation of MCAK controls ciliogenesis is interesting but limited. Have the authors investigated the effect of expressing physiological levels of a non-degradable MCAK mutant (unable to bind FBXW5) in cells?

As mentioned above, despite an intensive effort we were not able to identify amino acids that are required for the interaction between MCAK and Fbxw5. We could thus not investigate the effect of non-degradable MCAK mutants. We agree with the reviewer that such an experiment would be a very elegant way to further investigate the contribution of MCAK stabilisation on the ciliogenesis defect upon Fbxw5 depletion. However, as our new data (see response to reviewer #1) indicate that this ciliogenesis defect is probably due to a combined increase in the three different kinesin-13 proteins Kif2a, Kif2b and MCAK, we are not sure if such an approach is able to yield the expected results. Destroying the binding sites for all three kinesin-13 proteins would of course be an option, but even if we would be able to map these sites, expressing all these different mutants simultaneously to endogenous levels while simultaneously knocking down the wild type proteins would be extremely difficult.

We now adjusted our manuscript such that it better takes into account the three orthologs (title, abstract and discussion) and we hope that the reviewer agrees that we provide sufficient data to claim that the stabilisation of these proteins upon Fbxw5 depletion is at least to a large extent responsible for the ciliogenesis defect, especially considering the important rescue experiment on co-depletion of MCAK or Kif2a and Kif2b.

Minor points:

Figure 4A. To have a clearer picture of MCAK levels during cell cycle, the authors should synchronize, release from the synchronization block and collect cells at different time points. Such a time course would provide more information about the oscillation of MCAK during cell cycle that is only hinted in this figure.

We thank the reviewer for his/her suggestion, as we believe that this is indeed an important experiment, which is able to clarify the issue of high MCAK levels within the nocodazole arrest. We carried out double-thymidine block release experiments and collected samples within 14 hours post-release (Figure EV3A). Here, MCAK amounts were only moderately increased during mitosis (indicated by high CyclinB1 levels), suggesting that the stronger signals upon nocodazole treatment are probably due to a prolonged arrest. This is important, because it makes the Fbxw5-dependent regulation of MCAK during G₂/M much more meaningful if it does not coincide with a drastic increase in the amounts of the substrate.

The authors state: "MCAK amounts were slightly increased in asynchronously

growing cells", however no difference is detected in the levels of MCAK in figure 4A. I do not see correspondence between the levels of MCAK in asynchronous cells in the blot in figure 4A and the ones in graph 4B, even considering the ratio with the levels of the loading control UBA2. I understand that the graph represents the average of 4 independent experiments, but can the authors show a more representative blot?

We agree with the reviewer that the blot we've shown is not ideal and we replaced it with a more representative blot.

Figure 4A. The band corresponding to FBXW5 is barely visible. The authors should show a longer exposure FBXW5 blot in addition to the one already present.

Fbxw5 signals are better visible in the new blot.

Figure 3b. Do the "only Ubch5b" and "only Cdc34" samples contain also components of the SCF-FBXW5 complex? If so, specify it in the legend.

"Only Ubch5b" or "only Cdc34" referred to the E2 enzyme used, but this was indeed misleading. We now added a detailed description of components included or omitted within individual samples in the Figure.

Figure 1A. The calculated size of Cul1 size is about 89 kDa and it usually runs on SDS-PAGE at 70-80 kDa, however, in this figure it runs at about 50 kDa. Are they truncated forms? If so, explain.

In Figure 1, Cul1 was obtained via a split-and-coexpress method (Li et al., 2005), in which the C- and N-terminal domains are co-expressed as individual proteins and therefore run as two distinct bands at around 50 kDa. We added this explanation into the figure legend. For all other experiments, full length Cul1 from insect cells was used, which runs at around 85 kDa (Figure EV2A and B).

Figure 2C. Additional negative controls, such as for instance HT-FBXW7 (or any other FBXW) with MCAK-NL, are needed.

We repeated the experiment and included this time *HT-FBXW7* as an additional negative control, which displayed equal signals as the other negative controls.

Referee #3:

Review of SCFFbxw5 targets MCAK in G2/M to facilitate ciliogenesis in the following cell cycle by Schweiggert et al.

This manuscript identifies the microtubule destabilising kinesin, MCAK, as a substrate of the SCFFbxw5 E3 ligase. Briefly, in search of new SCFFbxw5 substrates the authors performed an in vitro ubiquitylation screen on a protein array, and isolated 161 candidate proteins that included known and new substrates such as MCAK. The report convincingly demonstrates that MCAK can be ubiquitylated by this E3 ligase and also that this modification predominantly occurs via insertion of K48-linked ubiquitin chains at multiple sites within MCAK.

Having isolated MCAK as a substrate of SCFFbxw5, the authors went on to investigate the purpose of this modification, and carried out a number of cell synchronisation, siRNA and protein overexpression studies to this end. Their key conclusion is that SCFFbxw5-dependent ubiquitylation of MCAK during G2 is important for ciliogenesis in G0. In particular, the authors find that depletion of SCFFbxw5 or exogenous expression of MCAK in serum-starved cells preclude cilia assembly. Co-depletion of MCAK and SCFFbxw5 restores normal ciliogenesis, indicating that in absence of MCAK, SCFFbxw5 is dispensable for cilia formation; this piece of data represents the most compelling evidence that SCFFbxw5-dependent degradation of MCAK promotes cilia assembly.

Overall, this manuscript is both interesting and informative. However, the conclusion that degradation of MCAK by SCFFbxw5 occurs in G2 with the outcome manifesting in ciliogenesis during the next cell cycle needs further experimental support.

We thank the reviewer for his/her positive reply and hope that our comments and the additional data provided in the revised version are sufficient to support our hypothesis on the G₂ regulation and its impact on ciliogenesis.

Specific points:

1. The impact of Fbxw5 depletion on MCAK levels is shown on a western blot in Fig 4A. Corresponding quantitation in Fig 4B compares signal intensities by normalising to the control of each condition. I do find it unusual to show the fold differences in the same graph when the controls are different. It may also be more informative to show the data normalised against Uba2 only, as this would enable readers to compare effects of different treatments. For example, the impact of nocodazole on MCAK levels is much higher than any other treatment, yet this is not apparent from the graph. The authors may also want to consider changing presentation of the graph in Fig 6B.

We normalized each sample to the control of each condition because comparing bands of highly different intensities (such as nocodazole vs serum starvation) is very inaccurate because it would have to be done on same exposure settings giving either a barely visible band (serum starvation) or an oversaturated one (nocodazole). We

therefore decided to now display each arrest in a separate bar chart (for both Figures) and hope that this is enough to avoid the impression that the controls are the same under each condition. As already mentioned above in response to reviewer #2, we also included a cell synchronisation experiment (Figure EV3A), in which the oscillation of MCAK during the cell cycle is better visible.

2. Fig.4C shows very clearly that centrosomal MCAK levels in serum-starved cells are affected by Fbxw5 depletion. It is a compelling idea that SCFFbxw5 may target a specific centrosomal pool of MCAK. A caveat here is that due to its multiple substrates SCFFbxw5 depletion is likely to have pleiotropic effects on the cell cycle and even in centrosomes, making it difficult to conclude that centrosomal increase in MCAK is solely due to depletion of this E3 ligase. The manuscript could be made much more impactful if lysine-mutant MCAK was generated that is resistant to SCFFbxw5. SCFFbxw5-resistant MCAK expressed as a transgene or a gene edited version would enable more specific functional studies.

As already mentioned above in the response to reviewer #2, we carried out mass spectrometry analysis of lysine residues modified within our *in vitro* ubiquitylation assay and identified 18 lysine residues that are modified by SCF^{Fbxw5} (Fig EV2M). This high number is actually in line with previous proteomics studies in cells and fits well to the multiple ubiquitylation species that are still observed when using methylated ubiquitin (which cannot form chains anymore and displays therefore only mono-ubiquitylation) in Figure 3G (last lane, top panel). We hope that the reviewer agrees that mutational disruption of so many lysine residues would almost certainly provoke other side effects and potentially even disrupt the structure, which would render the interpretation of such experiments very difficult.

Regarding pleiotropic (indirect) effects that could lead to the increase in centrosomal MCAK levels, we would like to point towards the wealth of evidence we have now collected to confidently claim that Fbxw5 is at least to a large part directly responsible for this increase, such as interaction between Fbxw5 and MCAK (at the endogenous level, with purified proteins and within intact cells (Figure 2)), the comprehensive characterisation of an efficient and specific ubiquitylation reaction *in vitro* affecting lysine residues that match well the ones identified in cell-based proteomic studies (Figure 3 and Figure EV2), stabilisation experiments (Figure 4C and Figure 7C) as well as CHX chase experiments in arrested and synchronised cells (Figure 6, see also below, response to Specific point #4).

3. It is not easy to reconcile findings that degradation of MCAK by SCFFbxw5 in G2 is required for ciliogenesis in the next G0. What makes this a difficult model is that MCAK levels peak in mitosis; this would suggest that the MCAK pool that is degraded in G2 would not be replenished by mitotic MCAK and that this pool is also inaccessible to APC/C. Certain centrosomal structures such as subdistal and distal appendages are removed/remodelled when cells enter mitosis, and could be candidates for such pools. The authors could investigate where exactly MCAK (endogenous and overexpressed, or upon Fbxw5 depletion) localises within the centrosome in G2, M and in G0 using super-resolution or expansion microscopy.

We agree that the high levels of MCAK under nocodazole arrest have been misleading. As mentioned above in our response to reviewer #2, we now included cell synchronisation experiments that reveal only a very modest increase in MCAK amounts during normal mitosis (Figure EV3A).

The question whether there is a stable pool of MCAK is indeed something that needed clarification - thank you very much for bringing this up. To address this, we used our mNG-MCAK expressing cells and conducted Fluorescence Recovery After Photobleaching (FRAP) experiments (Figure EV3E). As it turned out, the centrosomal pool of MCAK is highly dynamic, with recovery rates within the scale of a few seconds. This demonstrates that an increase in centrosomal MCAK could well be provoked by a general stabilisation of MCAK within the cytoplasm. This is actually what we believe is happening for the Fbxw5-dependent regulation, as Figure 5A and Figure EV4D and EV4E show a global increase in MCAK level within the whole population and not only at a specific localisation (within the IF images of Figure 4C the diffuse cytoplasmic signals are difficult to compare due to the low intensity). In line with that, using superresolution microscopy we could not detect a specific substructure of MCAK that is only present either within G₀ or G₂, or upon knock down of Fbxw5 (see Figure 5 for reviewers attention).

Figure 5 for reviewers attention. Stimulated emission depletion (STED) microscopy and image deconvolution. Cells were treated with the indicated siRNAs for 48 hours and either synchronised by double thymidine block and released for 6 hours (G₂) or serum starved for 24 hours (G₀). IF was carried out as before using anti-mouse Abberior® STAR 520SXP (MCAK) and anti-Rabbit Abberior® STAR 635P (ODF2) as secondary antibodies. STED imaging was performed on a Leica TCS SP8 3X STED system with a HC PLAPO 100×/1.40 NA STED White oil objective lens (Leica

Microsystems). The pinhole was set to 1 airy unit or smaller. Images were acquired using a white light laser at 515 nm excitation for STAR 520SXP and 635 nm for STAR 635P (abberior), respectively. For both dyes a 775 nm laser was used for STED. Signal detection was performed with HyD detectors (Leica Microsystems). Three dimensional (3D) confocal and STED image data at appropriate voxel sizes ($\leq 17 \times 17 \times 40$ nm XYZ) were acquired in sequential mode. The 3D image data set was deconvolved by Huygens Professional (SVI), and single slices from each 3D data set were used for presentation of STED microscopy results. Scale bar = 200 nm.

Regarding the point why the APC/C cannot compensate for the loss of Fbxw5, we would like to refer to our detailed response to reviewer #1, question #2.

4. Could the authors test if Fbxw5 localises to centrosome in G2 but not in G0? This would support the model propose. Also, can the authors exclude that the massive reduction in MCAK levels in Fbxw5-depleted and RO3306-treated cells is due to inhibition of CDK1 rather than the G2 arrest?

We investigated Fbxw5 at the endogenous level and upon overexpression using IF (Figure EV3B and C), but could not detect an obvious centrosomal localisation. We would like to refer again to the answer above (Specific Point #3).

The second aspect is indeed a very important one and we have conducted now cell synchronisation experiments in combination with CHX chase assays and siRNA treatment. Similar to RO-3306 treated cells, we observed an Fbxw5-dependent destabilisation of MCAK within a 6 hour release, which according to CyclinB1 levels corresponded to an enrichment of cells in G₂/M. This excludes side effects of the CDK1 inhibitor and further corroborates that the regulation is taking place in G₂. We thank the reviewer for this comment, as we believe that the new data significantly improved our study.

5. The authors should include western blots of mNG-MCAK overexpressing cells together with controls blotted with MCAK antibodies, so that readers can appreciate extent of overexpression. Immunofluorescence with MCAK antibodies would also be helpful.

This is a very good point that has also been brought up by reviewer #1, so that we would like to refer to our detailed answer above. We also included IF images with MCAK antibodies as we think that this is a good idea to better compare the levels within the cell (Figure EV4C and F). As mentioned above, we believe that after adjusting the ciliogenesis experiments, the extent of overexpression is now within a reasonable range for all overexpression experiments.

Thank you for submitting a revised version of your manuscript. Your study has now been seen by all original reviewers, who find that their main concerns have been addressed and now recommend publication of the manuscript after a minor revision. Therefore, I would like to invite you to address the following editorial issues before I can extend the official acceptance of the manuscript:

Referee #1:

The authors have addressed all my comments. The manuscript is now suitable for publication.

Referee #2:

The authors have addressed most of my concerns and strengthened the manuscript.

Referee #3:

The authors have satisfactorily addressed my comments, included additional experimental evidence and explained clearly where technical limitations prevented them from obtaining definitive data (i.e. non-degradable MCAK mutant).

I am however still puzzled about the model the authors propose that degradation of MCAK in G2 influences ciliogenesis. Results in the paper very clearly demonstrate that Fbxw5-dependent degradation of MCAK is important for ciliogenesis and also that Fbxw5 targets MCAK in G2. However, MCAK levels also increase when Fbxw5 is depleted in G0, and with fast-exchanging MCAK pools and MCAK levels peaking in mitosis, it remains unclear how Fbxw5-dependent removal of MCAK in G2 will impact on cilia growth in the next cell cycle. I would therefore suggest that the authors place less emphasis on this possibility, especially in their abstract.

Related to this point, the authors state in the abstract "In cells, SCFFbxw5 targets MCAK for proteasomal degradation specifically during G2/M." In the text they state "Interestingly, the most pronounced difference appeared in quiescent cells that had been serum-starved for 24 hours. Here, MCAK levels went almost below detection in control samples, but were 4-fold higher upon Fbxw5 knockdown." Clearly, the effect on MCAK is seen both in G2/M and G0, and perhaps the abstract could be refined to reflect these results more closely.

Corresponding Author Name: Niels Mailand

Journal Submitted to: The EMBO Journal

Manuscript Number: EMBOJ-2020-107413